# Residual Feature Integration is Sufficient to Prevent Negative Transfer

**Yichen Xu**[1*]  **Ryumei Nakada**[2*]  **Linjun Zhang**[3†]  **Lexin Li**[1]
[1]University of California, Berkeley
[2]Harvard University
[3]Rutgers University
{yichen_xu, lexinli}@berkeley.edu
lz412@stat.rutgers.edu
ryumei_nakada@hms.harvard.edu

## Abstract

Transfer learning has become a central paradigm in modern machine learning, yet it suffers from the long-standing problem of negative transfer, where leveraging source representations can harm rather than help performance on the target task. Although empirical remedies have been proposed, there remains little theoretical understanding of how to reliably avoid negative transfer. In this paper, we investigate a simple yet remarkably effective strategy: augmenting frozen, pretrained source-side features with a trainable target-side encoder that adapts target features to capture residual signals overlooked by models pretrained on the source data. We show this residual feature integration strategy is sufficient to provably prevent negative transfer, by establishing theoretical guarantees that it has no worse convergence rate than training from scratch under the informative class of target distributions up to logarithmic factors, and that the convergence rate can transition seamlessly from nonparametric to near-parametric when source representations are informative. To our knowledge, this is the first theoretical work that ensures protection against negative transfer. We carry out extensive numerical experiments across image, text and tabular benchmarks, and empirically verify that the method consistently safeguards performance under distribution shift, label noise, semantic perturbation, and class imbalance. We additionally demonstrate that this residual integration mechanism uniquely supports adapt-time multimodality extension, enabling a pretrained single-cell foundation model to incorporate spatial signals for lymph-node anatomical classification despite the source model being trained without them. Our study thus advances the theory of safe transfer learning, and provides a principled approach that is simple, robust, architecture-agnostic, and broadly applicable.

## 1 Introduction

Transfer learning provides a fundamental paradigm in modern machine learning, where knowledge acquired from one task (source domain) is leveraged to enhance performance on another related task (target domain). It encompasses a wide range of applications, from adapting models across different sources or domains, to distilling knowledge from large, pretrained models into smaller, task-specific models. Yet, a critical and persistent challenge is negative transfer: the phenomenon where transferring knowledge degrades performance compared to simply training on the target data from scratch. This issue, which arises from mismatches between source and target distributions, has been documented across numerous scenarios [34; 6; 28; 20; 49; 46; 40]. It is especially concerning in high-stakes applications such as healthcare, where transferring from broad datasets like ImageNet to medical imaging can be detrimental [38; 6]. Despite its prevalence, there remains little theoretical understanding of how to reliably avoid negative transfer.

---

[*]Equal contribution
[†]Corresponding author

In this article, we identify and validate a simple yet remarkably effective strategy that provably prevents negative transfer, i.e., augmenting frozen, pretrained source-side features with a trainable target-side encoder that adapts target features to capture residual signals overlooked by models pretrained on the source data. We call this strategy Residual Feature Integration (REFINE). Its implementation is straightforward: after obtaining the transferred representation $f_{\text{rep}}(x)$ from the source domain, instead of relying solely on $f_{\text{rep}}(x)$, we further introduce a residual connection with a trainable feature encoder $h(x)$ that is learned from the target domain. We then combine $f_{\text{rep}}(x)$ and $h(x)$, and fit a *shallow* neural network on the concatenated representation $(f_{\text{rep}}(x), h(x))$. Intuitively, while $f_{\text{rep}}(x)$ captures transferable features, it may omit target-specific signals that are critical for accurate prediction in the target domain. The residual connection via $h(x)$ compensates for this omission, ensuring that key information in the target domain is preserved. Furthermore, because $f_{\text{rep}}(x)$ already encodes a substantial portion of the predictive signal, learning from the joint representations $(f_{\text{rep}}(x), h(x))$ can potentially be achieved with a much simpler class of functions than learning from $x$ or $h(x)$ alone. We demonstrate, both theoretically and empirically, that this strategy is *sufficient to prevent negative transfer* across a broad range of settings.

Our contributions are threefold. First, we identify the residual connection, a widely adopted structural component originally devised to address optimization challenges in deep neural networks [15; 22], as a powerful mechanism for provably avoiding negative transfer. This strategy in turn offers a lightweight, robust, architecture-agnostic, and broadly applicable enhancement to transfer learning pipelines. We further identify an under-explored form of negative transfer that arises when source models lack modalities available only at adaptation time, and demonstrate that REFINE uniquely enables such adapt-time multi-modality extension on a single-cell foundation model for lymph-node domain classification. Second, we formally justify this simple yet remarkably effective approach through a rigorous theoretical analysis, which is the main contribution of this article. Specifically, we show that augmenting any frozen $f_{\text{rep}}$ with a trainable $h(x)$ guarantees that the resulting predictor achieves a convergence rate of prediction risk that is never worse than that obtained by training from scratch on the target data alone. In other words, REFINE is inherently robust against negative transfer in the worst-case scenario. Moreover, our prediction risk bound seamlessly transitions from a nonparametric convergence rate to a near-parametric rate when source representations are informative. Finally, we conduct extensive experiments on benchmark datasets spanning image, text, and tabular domains, and compare REFINE with multiple alternative solutions. We empirically verify that our method consistently mitigates negative transfer, especially under significant representational mismatch or task divergence.

## 2 Related Work

**Transfer learning**. Linear probing [26] and adapter-based feature extraction [18] are two of the most widely used transfer learning approaches. Both methods operate by extracting penultimate-layer features from a pretrained model in the source domain, followed by fine-tuning the final layer using data in the target domain. The main difference between the two is that linear probing employs a linear layer, while the adapter method uses a shallow neural network. Both are computationally efficient, but both are vulnerable to negative transfer. Knowledge distillation is another widely used transfer learning technique, where a large pretrained foundation model (the teacher) transfers knowledge to a simpler model (the student) that is typically fine-tuned in the target domain with substantially reduced complexity [16]. However, distillation remains vulnerable to negative transfer, especially when the teacher is poorly aligned with the target domain or when the transferred knowledge is too complex for the student to absorb effectively [10]. Our approach is applicable not only to knowledge transfer in foundation models, but also to general transfer learning settings.

**Negative transfer mitigation**. To mitigate negative transfer, various empirical remedies have been proposed, most of which focus on developing metrics that estimate similarity between source and target domains [11; 29; 47; 1]. Yet in practice, such similarity measures are often difficult to quantify, and sometimes require specialized loss functions or architectures, which limits their applicability [17]. [27] proposed SAFEW, which constructs an ensemble of source-domain models using a min–max framework. While theoretically sound, this method is computationally intensive and relies on the assumption that the optimal predictor can be expressed as a convex combination of source classifiers. [45] introduced DANN-GATE, a state-of-the-art solution that reduces negative transfer by combining adversarial training with a gating mechanism to filter out misleading source samples.

While practically effective, this method requires direct access to source data and is primarily empirical, lacking theoretical guarantees. In contrast, our method does not require access to original training data in the source domain and comes with rigorous theoretical guarantees. We also examine a largely overlooked form of negative transfer in which the source model lacks modalities that become available only at adaptation time. This setting is seldom discussed in multimodal learning [2], where it is typically assumed that all modalities are present during source-model training. Existing approaches cannot exploit such missing-modality information without retraining on source data. REFINE uniquely enables adapt-time multimodality extension without access to source data.

**Residual learning, stacking, and parameter-efficient fine-tuning**. Several methods are conceptually related to REFINE, although they do not explicitly target negative transfer in transfer learning. Residual learning, a core idea in architectures such as ResNet [15] and algorithms like gradient boosting [22], was originally developed to ease optimization challenges or improve prediction. Its potential for addressing negative transfer, however, remains unexplored. Stacking is an ensemble technique that combines predictions from multiple base models through a meta-learner trained on validation outputs. This approach is generally more robust than simple model averaging [4], but it assumes that all external models are reliable [9; 12], and requires aligned output spaces, which restricts its applicability across different types of tasks. Parameter-efficient fine-tuning methods, such as LoRA [19], insert lightweight, trainable modules into pretrained models to enable domain adaptation without modifying the original weights. Such approaches are effective and significantly reduces parameter costs, but struggles when source representations misalign with the target domain. Besides, it requires access to pretrained model weights and computational graphs, limiting their flexibility, particularly in the multi-source transfer setting.

## 3 PROBLEM FORMULATION AND ALGORITHM

Transfer learning aims to leverage knowledge from a source task to improve performance on a related target task. A common practice is to use a representation function $f_{\text{rep}}$ learned from a large source dataset $D^s$ under a source distribution $\mathbb{P}^s$ as an extracted feature for the target task. However, if $f_{\text{rep}}$ does not align well with the target distribution $\mathbb{P}^t$, naively reusing it can lead to negative transfer, resulting in degraded performance compared to using the target data alone.

We formalize the Residual Feature Integration (REFINE) approach. The objective is to construct a method such that, when $f_{\text{rep}}$ aligns well with the target distribution, we effectively leverage transferred knowledge and outperform models trained from scratch on target data only, and when $f_{\text{rep}}$ misaligns with the target distribution, safeguard against negative transfer and outperform models that rely solely on $f_{\text{rep}}(x)$. We focus on the supervised learning task. Let $D^t = \{(x_i, y_i)\}_{i=1}^n \sim \mathbb{P}^t$ denote the labeled dataset from the target task. Assume access to a frozen extracted feature $f_{\text{rep}} : \mathcal{X} \to \mathbb{R}^p$ trained on an external source data $D^s$. Define a class $\mathcal{H}$ of trainable feature encoders $h : \mathcal{X} \to \mathbb{R}^q$ and a class $\mathcal{W}$ of trainable adapters $w : \mathbb{R}^{p+q} \to \mathbb{R}^k$ on top of

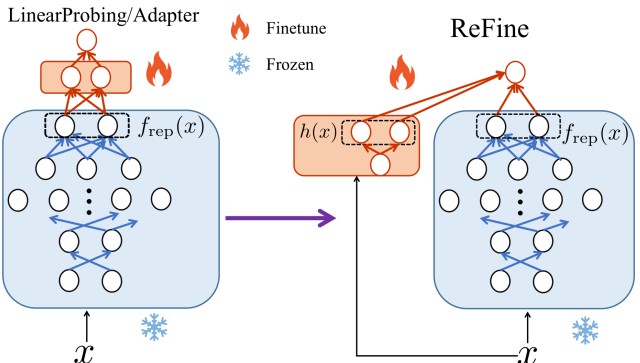

Figure 1: A schematic overview of REFINE.

$(f_{\text{rep}}(x), h(x))$. Let $\hat{w}_{\text{ft}}$ be the trained adapter on top of the baseline model, and let $\hat{g}_{\text{sc}}$ be the model trained from scratch on $x$. We seek to learn both the encoder $\hat{h}$ and the adapter $\hat{w}$, such that the expected excess risk of $\hat{w} \circ (f_{\text{rep}}, \hat{h})$ over the target distribution is bounded by the minimum of the excess risks of the two baselines: $\hat{w}_{\text{ft}} \circ f_{\text{rep}}$ and $\hat{g}_{\text{sc}}$.

Algorithm 1 outlines the REFINE approach. It extracts $f_{\text{rep}}(x)$ from the penultimate layer of a frozen pretrained model, and combines it with the residual connection $h(x)$. The concatenated features

$(f_{\text{rep}}(x), h(x))$ are passed to a linear classifier for prediction, where only $h(x)$ and the adapter $w$ are updated, whereas the pretrained model and $f_{\text{rep}}(x)$ remain unchanged. This design allows REFINE to efficiently complement transferred knowledge with adapted features from the target data, and thus recover potentially lost information during the forward pass in the frozen source model. Figure 1 gives a schematic overview of REFINE.

## 4 THEORETICAL ANALYSIS

We provide a theoretical analysis to prove that REFINE is robust to negative transfer. The intuition and core insight is that the residual connection provides a natural transition: if the external representation $f_{\text{rep}}$ is uninformative, the residual network $h$ can still learn the target function from the raw input, recovering the performance of training from scratch. Conversely, if $f_{\text{rep}}$ is informative, $h$ only needs to learn the simpler residual function, reducing the effective complexity of the problem and accelerating the learning. This intuition is formalized in two ways: a no-negative-transfer guarantee showing that, under mild growth conditions on model capacity, REFINE is never worse than either training from scratch or using a linear probe on $f_{\text{rep}}$, and a risk bound showing that its convergence rate smoothly interpolates between the standard nonparametric rate and a near-parametric rate depending on the quality of the external representation.

We formalize this intuition within the framework of nonparametric regression. We consider the model with a *trainable* residual feature encoder $h$:
$$g(x) = uh(x) + v^\top f_{\text{rep}}(x),$$
where $h(x)$ is a (clipped) ReLU network over raw input, combined with a linear probe on the feature $f_{\text{rep}}(x)$. We establish the risk bound demonstrating that, for moderate capacity of $h$, REFINE's excess risk is no worse than the excess risk of the model trained from scratch or the linear probe on $f_{\text{rep}}(x)$. Furthermore, when the capacity of $h$ is tuned to the difficulty of the residual task, the rate adapts and improves, showcasing its ability to effectively leverage useful prior information from $f_{\text{rep}}(x)$.

**Formal Setup.** We consider the nonparametric regression setup adopted in the statistical analysis of deep neural networks [42; 39; 24]. Specifically, we observe $n$ i.i.d. pairs $(X_i, Y_i)_{i \in [n]} \sim \mathbb{P}^t$ with support on $[0,1]^d \times \mathbb{R}$ following the model
$$Y_i = f^*(X_i) + \epsilon_i, \tag{1}$$
where $f^* : [0,1]^d \to [-1,1]$ is the ground-truth regression function, $(X_i)_{i \in [n]}$ are i.i.d. samples from the marginal distribution $\mathbb{P}^t_X$ on $X$, and $(\epsilon_i)_{i \in [n]}$ are i.i.d. Gaussian with variance $\sigma^2 = \Theta(1)$, independent of $(X_i)_{i \in [n]}$. We assume $\mathbb{P}^t_X$ admits a positive continuous density on $[0,1]^d$ upper bounded by an absolute constant. Under this set-up, the expected loss for a given function $g$ is $\mathcal{R}_{\mathbb{P}^t}(g) = \mathbb{E}_{(X,Y) \sim \mathbb{P}^t}[(g(X) - Y)^2]$.

To facilitate the theoretical analysis, following the standard setup of nonparametric regression, we consider $f^*$ to be Hölder smooth. Specifically, for a non-integer $\beta > 0$, the Hölder norm for $f^*$ that are $\lfloor \beta \rfloor$-times differentiable on $[0,1]^d$ is

$$\|f\|_{\mathcal{C}^\beta} := \max\left\{ \max_{a \in \mathbb{N}^d : \|a\|_1 \le \lfloor\beta\rfloor} \sup_{x \in [0,1]^d} |\partial^a f(x)|, \max_{a \in \mathbb{N}^d : \|a\|_1 = \lfloor\beta\rfloor} \sup_{x \ne x'} \frac{|\partial^a f(x) - \partial^a f(x')|}{\|x - x'\|^{\beta - \lfloor\beta\rfloor}} \right\}.$$

---

**Algorithm 1** The residual feature integration (REFINE) method.

1: **Input:** Training data $\mathcal{D}_{\text{train}} = (X_i, Y_i)_i$, test data $\mathcal{D}_{\text{test}}$, pretrained model $f$, loss function $\ell$.
2: **Output:** Prediction of the label $\hat{y}(x_0)$ for $x_0 \in \mathcal{D}_{\text{test}}$.
3: **Training Phase:**
4:     (a) Extract $f_{\text{rep}}(x)$ from the penultimate-layer of a frozen pretrained model $f$.
5:     (b) Construct the concatenated features $C_h(x) := (f_{\text{rep}}(x), h(x))$.
6:     (c) Let $(\hat{w}, \hat{h})$ be the minimizer of $\sum_i \ell(w(C_h(X_i)), Y_i)$ while freezing $f_{\text{rep}}$.
7: **Prediction Phase:**
8:     (a) Compute $C_h(x_0)$ with the frozen $f$.
9:     (b) Obtain the final prediction $\hat{y}(x_0)$ based on $\hat{w}(C_{\hat{h}}(x_0))$.

---

The unit ball is $\mathcal{C}_{\mathrm{u}}^{\beta} := \{f : [0,1]^d \to \mathbb{R} : f \text{ is } \lfloor \beta \rfloor\text{-times differentiable and } \|f\|_{\mathcal{C}^{\beta}} \leq 1\}$.

Further, we assume the residual connection $h : \mathbb{R}^d \to \mathbb{R}$ is realized by a ReLU network with width at most $W$, depth at most $L$, and weight magnitude at most $B$:

$$h(x) = A_{L'} x^{(L'-1)} + b_{L'}, \quad x^{(\ell)} = \sigma(A_\ell x^{(\ell-1)} + b_\ell) \ (\ell \in [L'-1]), \quad x^{(0)} = x, \quad (2)$$

for some $L' \leq L$, where $d_0 = d$, $d_{L'} = 1$, and $d_\ell \leq W$. Here $\sigma(z) = \max\{0, z\}$ is applied element-wise, $A_\ell \in [-B, B]^{d_\ell \times d_{\ell-1}}$, and $b_\ell \in [-B, B]^{d_\ell}$. The class is $\mathcal{H}_d(W, L, B)$, and we use its clipped counterpart $\bar{\mathcal{H}}_d(W, L, B) := \{x \mapsto \min\{1, \max\{-1, h(x)\}\} : h \in \mathcal{H}_d(W, L, B)\}$.

**Empirical risk minimization for REFINE.** We consider squared loss $\ell(y, y') = (y - y')^2$. Let $f_{\mathrm{rep}} : [0,1]^d \to \mathcal{B}_p(1)$ be an external representation with $\mathcal{B}_p(R) = \{u \in \mathbb{R}^p : \|u\| \leq R\}$. Define the REFINE class

$$\mathcal{G}_{d,p}(W, L, B; f_{\mathrm{rep}}) = \left\{ g : [0,1]^d \to \mathbb{R} \ \Big| \ g(x) = v^\top f_{\mathrm{rep}}(x) + uh(x), \ |u| \leq 1, \ \|v\| \leq 1, \ h \in \bar{\mathcal{H}}_d(W, L, B) \right\}.$$

We train $\hat{g}$ via empirical risk minimization,

$$\hat{g} = \underset{g \in \mathcal{G}_{d,p}(W, L, B; f_{\mathrm{rep}})}{\arg \min} \frac{1}{n} \sum_{i \in [n]} \ell(g(X_i), Y_i). \quad (3)$$

The effectiveness of REFINE depends on the quality of $f_{\mathrm{rep}}$. We quantify this by defining the best possible linear probe and the corresponding residual. Specifically, for any $f_{\mathrm{rep}} : [0,1]^d \to \mathcal{B}_p(1)$, the best linear probe is defined as

$$v^* = \underset{v \in \mathbb{R}^p}{\arg \min} \, \mathbb{E}\left[\{v^\top f_{\mathrm{rep}}(X_1) - f^*(X_1)\}^2\right].$$

The difficulty of learning the residual is then captured by its Hölder norm, which we denote as $\rho^* := \|v^{*\top} f_{\mathrm{rep}} - f^*\|_{\mathcal{C}^{\beta}}$. A small $\rho^*$ indicates that $f_{\mathrm{rep}}$ is highly informative for the target task.

We state a theorem that provides an upper bound on the generalization error of the empirical risk minimizer in equation 3, when the model capacity is chosen appropriately.

**Theorem 4.1** (Generalization Error of REFINE). *Assume $v^{*\top} f_{\mathrm{rep}} - f^* \in \mathcal{C}^{\beta}$. Let $\rho \geq 0$ be a tuning parameter, which serves as a proxy for the residual norm, and choose the network parameters for $h$ as*

$$L = c_1, \qquad W = c_2 \max\{n^{d/(2\beta+d)} \rho^{2d/(2\beta+d)}, 1\}, \qquad B = (\rho^* \vee 1) \max\{n\rho^2, 1\}^{c_3}, \quad (4)$$

*where $c_1, c_2, c_3 > 0$ depend on $\beta$, $d$ and $\gamma$. Let $\hat{g}$ be the empirical risk minimizer in (3) with the parameter specified as in (4). Then there exists $C > 0$, which depends on $\beta, d$, such that*

$$\mathbb{E}[\mathcal{R}_{\mathbb{P}'}(\hat{g}) - \mathcal{R}_{\mathbb{P}'}(f^*)] \leq C\left\{ \left(\rho^{2d/(2\beta+d)} \log n + \rho^{*2} \rho^{-4\beta/(2\beta+d)}\right) n^{-2\beta/(2\beta+d)} + \frac{p \log n}{n} \right\}. \quad (5)$$

The bound in (5) splits into a parametric term $p \log n / n$ for learning $v^*$ on top of $f_{\mathrm{rep}}$, and a non-parametric term with the standard minimax rate $n^{-2\beta/(2\beta+d)}$ for learning the residual modulated by the tuning parameter $\rho$ and the residual difficulty $\rho^*$. The tuning radius $\rho$ controls the effective capacity of $h$ via $W$ and $B$ in (4). That is, a larger $\rho$ increases the approximation power, achieving a smaller bias, but worsens the estimation, resulting in a larger variance factor $\rho^{2d/(2\beta+d)}$. On the other hand, a smaller $\rho$ regularizes $h$, which is preferable when the residual is genuinely small.

**Proof sketch of Theorem 4.1** For any $v$, decompose

$$f^*(x) = \underbrace{f^*(x) - v^\top f_{\mathrm{rep}}(x)}_{\text{residual}} + \underbrace{v^\top f_{\mathrm{rep}}(x)}_{\text{linear in } f_{\mathrm{rep}}(x)} \ .$$

The first term is fit by $h$ and the second by a linear probe on $f_{\mathrm{rep}}$. Approximation results for ReLU networks over $\mathcal{C}^{\beta}$ functions give the residual term at rate $n^{-2\beta/(2\beta+d)}$ with a capacity-dependent multiplier governed by $\rho$. A standard linear estimation yields the $p/n$ term for $v$. Choosing $(W, L, B)$ as in (4) implements this bias-variance trade-off. The full proof is deferred to Appendix A.

We remark that our theoretical results are derived under the squared-loss objective, following a long line of work that analyzes classification problems through regression surrogates [14; 50]. This approach aligns with common practice in the machine learning theory community, where regression surrogates are employed to derive insights for classification algorithms.

We further discuss two direct implications of Theorem 4.1.

**Corollary 4.2** (Fixed $\rho$). *Under the same conditions as in Theorem 4.1, for any fixed choice of $\rho > 0$, the bound in (5) implies that*

$$\mathbb{E}[\mathcal{R}_{\mathbb{P}^t}(\hat{g}) - \mathcal{R}_{\mathbb{P}^t}(f^*)] = \tilde{O}\Big(n^{-2\beta/(2\beta+d)} + \frac{p}{n}\Big).$$

This corollary indicates that, by introducing an additional residual connection $h$, REFINE never has a worse rate than $n^{-2\beta/(2\beta+d)}$ for fixed $p$, which is the standard minimax-optimal rate when training from scratch on $(X_i, Y_i)_{i\in[n]}$ for $\beta$-Hölder $f^*$ (See, for example, Theorem 3.2 in Györfi et al. [13]).

**Corollary 4.3** (Tuned $\rho$). *Under the same conditions as in Theorem 4.1, balancing (5) by choosing $\rho \downarrow \rho^*$ yields*

$$\mathbb{E}[\mathcal{R}_{\mathbb{P}^t}(\hat{g}) - \mathcal{R}_{\mathbb{P}^t}(f^*)] = \tilde{O}\Big(\rho^{*2d/(2\beta+d)}n^{-2\beta/(2\beta+d)} + \frac{p}{n}\Big). \tag{6}$$

This corollary indicates that, when $f_{\text{rep}}$ is well aligned with the target, i.e., a small $\rho^*$, choosing $\rho \asymp \rho^*$ effectively regularizes the residual network $h$ via the parameter choice in (4), which shrinks the nonparametric term so that the bound is dominated by the near-parametric $p/n$ term. Conversely, when $f_{\text{rep}}$ is misaligned, i.e., a large $\rho^*$, the nonparametric component dominates and the rate reverts to the classical $\beta$-Hölder minimax rate $n^{-2\beta/(2\beta+d)}$.

We now provide a corollary for no-negative-transfer guarantee. The key idea is to define a class of functions that can be approximated in the $\beta$-Hölder norm by a linear combination of $f_{\text{rep}}$ up to an error $\gamma > 0$:

$$\mathcal{F}^\beta(f_{\text{rep}}, \gamma) := \{f^* : [0,1]^d \to \mathbb{R}\big| \min_{v:\|v\|\leq 1} \|v^\top f_{\text{rep}} - f^*\|_{\mathcal{C}^\beta} \leq \gamma\}.$$

This class captures the functions that can be learned from the residual connection $h$ and the linear probe on $f_{\text{rep}}$, and thus serves as a target for the empirical risk minimization in equation 3. In a special case where $f_{\text{rep}}$ is not informative, i.e., $f_{\text{rep}} = 0$, the class reduces to the standard Hölder ball with radius $\gamma$: $\{f^*|\|f^*\|_{\mathcal{C}^\beta} \leq \gamma\}$.

**Corollary 4.4** (No-negative-transfer guarantee). *Fix $d, p \in \mathbb{N}_+$ and $\beta > 0$. Also fix $f_{rep} : [0,1]^d \to \mathbb{R}^p$ satisfying $v^\top f_{rep} \in \mathcal{C}_u^\beta$ for any unit vector $v \in \mathbb{S}^{p-1}$. Consider the model trained from scratch and the linear probe on $f_{rep}$ with comparable capacity:*

$$\hat{g}_{sc} = \underset{g\in\bar{\mathcal{H}}_d(W,L,B)}{\arg\min} \frac{1}{n}\sum_{i\in[n]} \ell(g(X_i), Y_i), \quad \hat{w}_{ft} = \underset{w\in\mathbb{R}^p}{\arg\min} \frac{1}{n}\sum_{i\in[n]} \ell(w^\top f_{rep}(X_i), Y_i).$$

*Then,*

$$\sup_{f^*\in\mathcal{F}^\beta(f_{rep},\gamma)} \mathbb{E}[\mathcal{R}_{\mathbb{P}^t}(\hat{g}) - \mathcal{R}_{\mathbb{P}^t}(f^*)]$$

$$= \tilde{O}\left(\min\left\{\sup_{f^*\in\mathcal{F}^\beta(f_{rep},\gamma)}\mathbb{E}[\mathcal{R}_{\mathbb{P}^t}(\hat{g}_{sc}) - \mathcal{R}_{\mathbb{P}}(f^*)], \sup_{f^*\in\mathcal{F}^\beta(f_{rep},\gamma)}\mathbb{E}[\mathcal{R}_{\mathbb{P}^t}(\hat{w}_{ft}^\top f_{rep}) - \mathcal{R}_{\mathbb{P}^t}(f^*)]\right\}\right)$$

*holds for any $\gamma \in [0, 1)$.*

Specifically, when $\gamma = 0$, i.e., when $f^*$ lies exactly in the linear span of $f_{\text{rep}}$, the excess risk of RE-FINE is, up to logarithmic factors, no worse than that of the linear probe on $f_{\text{rep}}$. When $\gamma > 0$, RE-FINE attains the standard nonparametric rate for estimating $\beta$-Hölder functions; this rate improves upon that of the linear probe on $f_{\text{rep}}$, which suffers from bias due to representational misalignment. Hence REFINE provably avoids negative transfer for this class of target functions.

We also provide the asymptotic no-negative-transfer guarantee for a *fixed $f^*$* under *any* mild model capacity in Appendix A.3.

## 5 NUMERICAL EXPERIMENTS

### 5.1 EXPERIMENT SETUP

We demonstrate that REFINE consistently mitigates negative transfer through extensive numerical experiments across image, text, and tabular modalities, using benchmark datasets including CIFAR-10, CIFAR-100 [25], STL [5], Clipart, Sketch [36], USPS, MNIST, Books, Kitchen, DVD, and Electronics [3]. We evaluate performance using classification accuracy, area under ROC (AUC), F1 score, and minimum class accuracy.

We also compare REFINE with a number of alternative solutions. In particular, NoTrans serves as a no-transfer baseline, reusing pretrained features without any adaptation. LinearProbe [26] trains only a linear classifier on top of frozen features, offering a lightweight baseline. Adapter [18] inserts a small trainable module into pretrained models, enabling efficient adaptation with limited parameters. Distillation [16] transfers knowledge from a frozen teacher to a student model through a combination of hard labels and soft predictions. LoRA [19] applies low-rank adaptations to weight matrices, achieving parameter-efficient fine-tuning. DANN-Gate [45] combines adversarial training with gating to encourage domain-invariant representations.

We consider a variety of experimental settings. In Section 5.2, we evaluate REFINE on datasets exhibiting natural distribution shift. In Section 5.3, we construct challenging scenarios to stress-test robustness under controlled perturbations. In Section 5.4, we examine an adapt-time multimodality extension setting based on spatial transcriptomics, where a new modality becomes available only after pretraining. Finally, in the Appendix, we include additional studies on source-free multi-source transfer (Appendix C.1) and tabular benchmark evaluations (Appendix C.4).

In our implementations, we train all models using stochastic gradient descent with a learning rate 0.01 and momentum 0.9, with pretraining for 60 epochs and fine-tuning for 30 epochs. We consider both CNNs and transformer architectures for the pretrained model $f_{\text{rep}}$ and the encoders $h$. We also carry out an ablation study in Appendix C.5 regarding the complexity of the encoder $h$, showing that REFINE remains effective across different choices of the model parameters for $h$.

We provide more details about the experiment setup and implementations in Appendix D.

### 5.2 SINGLE-SOURCE TRANSFER WITH NATURAL DISTRIBUTION SHIFT

In the first experiment setting, we evaluate REFINE on datasets that exhibit natural distribution shift. To provide a comprehensive assessment, we consider transfer tasks spanning both image and language, thereby covering cross-domain as well as cross-modality adaptation. For image, we include CIFAR-10, CIFAR-100, and STL-10, which offer complementary object recognition tasks with varying class granularity and image resolution. We further incorporate artistic domains, specifically, Clipart and Sketch, to capture substantial stylistic diversity, along with digit recognition benchmarks, USPS and MNIST, which provide structured and well-curated handwritten digits. For text, we adopt the datasets, Books, DVD, Electronics, and Kitchen, which span heterogeneous product categories and exhibit rich linguistic variations. We process the image datasets using convolutional neural networks (CNNs), and process the text datasets using transformers. This design allows us to assess transfer across distribution and domain shifts, and also under cross-modality and cross-model settings. Collectively, these datasets constitute a broad and rigorous benchmark for evaluating transfer learning methods.

We use the notation $A \rightarrow B$ to denote transfer learning from source domain $A$ to target domain $B$. Our evaluation covers diverse scenarios. Specifically, CIFAR100$\rightarrow$10 and CIFAR10$\rightarrow$100 test transfers across datasets with overlapping but non-identical class spaces and label granularity; CIFAR10$\rightarrow$STL reflects natural distribution shift due to resolution and dataset construction; Clipart$\rightarrow$Sketch represents cross-style adaptation between artistic domains; USPS$\rightarrow$MNIST examines digit recognition under handwriting and design difference; and Books$\rightarrow$Kitchen and DVD$\rightarrow$Electronics capture cross-topic sentiment transfer, where vocabulary and linguistic style vary considerably. We exclude knowledge distillation [16] in this comparison, as it requires identical class spaces across source and target, which do not apply here.

| Dataset | Method | Accuracy | AUC | F1 | Min CAcc |
|---|---|---|---|---|---|
| CIFAR100→10 | NoTrans | **56.5820 ± 0.3659** | **0.9005 ± 0.0012** | **0.5634 ± 0.0046** | **37.2000 ± 3.4117** |
| | LinearProb | 38.9260 ± 0.5463 | 0.8284 ± 0.0017 | 0.3815 ± 0.0051 | 16.9400 ± 3.7441 |
| | Adapter | 38.2320 ± 0.3111 | 0.8247 ± 0.0016 | 0.3754 ± 0.0071 | 16.4600 ± 5.4544 |
| | LoRA | 43.1360 ± 0.3239 | 0.8603 ± 0.0003 | 0.4237 ± 0.0046 | 20.1400 ± 4.1020 |
| | DANN-Gate | 43.2220 ± 0.1295 | 0.8605 ± 0.0005 | 0.4214 ± 0.0040 | 17.4800 ± 4.7755 |
| | REFINE | 54.4000 ± 0.3336 | 0.8942 ± 0.0026 | 0.5406 ± 0.0051 | 33.6200 ± 2.8273 |
| CIFAR10→100 | NoTrans | 18.3200 ± 0.5254 | 0.8140 ± 0.0050 | 0.1774 ± 0.0052 | 1.0000 ± 0.8944 |
| | LinearProbe | 7.0140 ± 0.3347 | 0.7489 ± 0.0011 | 0.0496 ± 0.0034 | 0.0000 ± 0.0000 |
| | Adapter | 6.5640 ± 0.2875 | 0.7499 ± 0.0008 | 0.0459 ± 0.0026 | 0.0000 ± 0.0000 |
| | LoRA | 6.8240 ± 0.1037 | 0.7558 ± 0.0010 | 0.0463 ± 0.0015 | 0.0000 ± 0.0000 |
| | DANN-Gate | 5.1980 ± 0.3924 | 0.7341 ± 0.0055 | 0.0285 ± 0.0033 | 0.0000 ± 0.0000 |
| | REFINE | **18.5880 ± 0.5494** | **0.8276 ± 0.0053** | **0.1787 ± 0.0057** | **1.4000 ± 0.8000** |
| CIFAR10→STL | NoTrans | 48.6925 ± 0.6338 | 0.8683 ± 0.0032 | 0.4831 ± 0.0089 | **26.8000 ± 4.9006** |
| | LinearProbe | 50.2725 ± 0.3016 | 0.8795 ± 0.0015 | 0.4955 ± 0.0067 | 18.9250 ± 6.1546 |
| | Adapter | 49.2900 ± 0.7344 | 0.8773 ± 0.0008 | 0.4865 ± 0.0096 | 15.6750 ± 6.6340 |
| | LoRA | 50.7550 ± 0.3793 | 0.8813 ± 0.0016 | 0.4930 ± 0.0040 | 5.6750 ± 2.6933 |
| | DANN-Gate | 47.7050 ± 0.6586 | 0.8659 ± 0.0013 | 0.4712 ± 0.0104 | 13.9250 ± 5.3424 |
| | REFINE | **53.4175 ± 0.3628** | **0.8944 ± 0.0013** | **0.5301 ± 0.0053** | 25.9750 ± 3.5693 |
| Clipart→Sketch | NoTrans | 18.8804 ± 1.3709 | 0.7170 ± 0.0117 | 0.1828 ± 0.0119 | 0.0000 ± 0.0000 |
| | LinearProbe | 18.3430 ± 0.8649 | 0.7290 ± 0.0065 | 0.1727 ± 0.0087 | 0.0000 ± 0.0000 |
| | Adapter | 18.2356 ± 0.5807 | **0.7369 ± 0.0059** | 0.1549 ± 0.0040 | 0.0000 ± 0.0000 |
| | LoRA | 16.9010 ± 0.6906 | 0.6937 ± 0.0043 | 0.1671 ± 0.0069 | 0.0000 ± 0.0000 |
| | DANN-Gate | 16.5786 ± 0.4868 | 0.6942 ± 0.0021 | 0.1544 ± 0.0048 | 0.0000 ± 0.0000 |
| | REFINE | **20.3403 ± 0.4768** | 0.7338 ± 0.0043 | **0.1968 ± 0.0059** | 0.5263 ± 1.0526 |
| USPS→MNIST | NoTrans | 62.0740 ± 8.7771 | 0.9566 ± 0.0073 | 0.5967 ± 0.0969 | 9.2863 ± 12.1512 |
| | LinearProbe | 66.9960 ± 1.0095 | 0.9469 ± 0.0050 | 0.6563 ± 0.0086 | 9.1576 ± 3.5478 |
| | Adapter | 61.8660 ± 3.0334 | 0.9375 ± 0.0085 | 0.5952 ± 0.0441 | 8.8750 ± 7.2427 |
| | LoRA | 64.8240 ± 0.8520 | 0.9333 ± 0.0045 | 0.6435 ± 0.0135 | 29.3265 ± 13.5652 |
| | DANN-Gate | 52.2080 ± 3.6669 | 0.9012 ± 0.0185 | 0.4853 ± 0.0482 | 0.0198 ± 0.0396 |
| | REFINE | **70.0460 ± 2.1721** | **0.9582 ± 0.0053** | **0.6954 ± 0.0194** | **31.6157 ± 14.5527** |
| Books→Kitchen | NoTrans | 71.6600 ± 1.3632 | 0.7848 ± 0.0155 | 0.7161 ± 0.0137 | **68.6000 ± 2.9719** |
| | LinearProbe | 66.7400 ± 3.1455 | 0.7568 ± 0.0278 | 0.6571 ± 0.0401 | 51.5600 ± 9.7336 |
| | Adapter | 71.3400 ± 0.1356 | 0.7839 ± 0.0008 | 0.7111 ± 0.0015 | 62.8800 ± 2.9027 |
| | LoRA | 66.9600 ± 0.2154 | 0.7279 ± 0.0018 | 0.6695 ± 0.0022 | 65.6400 ± 0.4079 |
| | DANN-Gate | 66.6000 ± 0.0894 | 0.7330 ± 0.0006 | 0.6659 ± 0.0009 | 64.6800 ± 0.6997 |
| | REFINE | **72.7200 ± 1.6522** | **0.8147 ± 0.0133** | **0.7248 ± 0.0189** | 65.5200 ± 6.4778 |
| DVD→Electronics | NoTrans | 68.5200 ± 2.8979 | 0.7585 ± 0.0304 | 0.6806 ± 0.0338 | 59.8000 ± 9.8298 |
| | LinearProbe | 66.0600 ± 0.5122 | 0.7266 ± 0.0017 | 0.6580 ± 0.0072 | 58.3600 ± 4.5579 |
| | Adapter | 65.8600 ± 0.3200 | 0.7206 ± 0.0008 | 0.6577 ± 0.0037 | 61.4400 ± 2.5935 |
| | LoRA | 66.5600 ± 0.3555 | 0.7170 ± 0.0013 | 0.6656 ± 0.0036 | **65.4000 ± 0.4899** |
| | DANN-Gate | 66.9000 ± 0.1897 | 0.7196 ± 0.0013 | 0.6686 ± 0.0019 | 63.5600 ± 0.2653 |
| | REFINE | **70.3400 ± 0.9972** | **0.7886 ± 0.0115** | **0.6995 ± 0.0122** | 61.7200 ± 7.5181 |

Table 1: Single-source transfer learning with natural distribution shift.

Table 1 reports the results. REFINE consistently achieves competitive or superior performance compared to alternative methods across all scenarios. On transfers with large label-space difference, including CIFAR100→10 and CIFAR10→100, REFINE improves accuracy by over $10 - 15\%$ relative to Adapter, LoRA, and DANN-Gate, substantially narrowing the gap to the no-transfer baseline while remaining robust to negative transfer. On transfers under natural resolution or stylistic shift, including CIFAR10→STL, Clipart→Sketch, REFINE achieves $3 - 4\%$ accuracy gains over the strongest alternative, along with consistent improvements in AUC and F1. On transfers with digit benchmarks, including USPS→MNIST, it yields $5 - 10\%$ accuracy gains, and much higher minimum class accuracy, indicating stronger preservation of performance on underrepresented classes. On transfers with cross-topics, including Books→Kitchen, DVD→Electronics, REFINE delivers $2 - 4\%$ improvements across all metrics. Overall, REFINE not only avoids the severe degradation observed in other methods, but also provides reliable accuracy lifts of $5 - 15\%$ across image and text domains under diverse settings of distribution shifts.

## 5.3 SINGLE-SOURCE TRANSFER UNDER LABEL NOISE, SEMANTIC PERTURBATION, AND CLASS IMBALANCE

In the second experiment setting, we deliberately construct challenging scenarios to stress-test various transfer learning methods. Using CIFAR-10 with CNNs, we introduce four types of challenges

| Dataset | Setting | Method | Acc | AUC | F1 | MinCAcc |
|---------|---------|--------|-----|-----|----|---------|
| CIFAR-10 | 40% flips | NoTrans | $56.05 \pm 0.64$ | $0.9037 \pm 0.0028$ | $0.5580 \pm 0.0080$ | $32.40 \pm 5.84$ |
| | | LinearProbe | $65.54 \pm 0.06$ | $0.9378 \pm 0.0003$ | $0.6561 \pm 0.0008$ | $42.82 \pm 1.45$ |
| | | Adapter | $65.78 \pm 0.19$ | $0.9376 \pm 0.0007$ | $0.6581 \pm 0.0024$ | $\mathbf{45.20 \pm 2.29}$ |
| | | Distill | $57.01 \pm 0.58$ | $0.9115 \pm 0.0016$ | $0.5674 \pm 0.0032$ | $34.84 \pm 4.53$ |
| | | LoRA | $65.47 \pm 0.12$ | $0.9374 \pm 0.0004$ | $0.6545 \pm 0.0018$ | $42.38 \pm 0.89$ |
| | | DANN-Gate | $65.40 \pm 0.15$ | $0.9353 \pm 0.0006$ | $0.6539 \pm 0.0016$ | $43.40 \pm 2.22$ |
| | | REFINE | $\mathbf{66.23 \pm 0.32}$ | $\mathbf{0.9388 \pm 0.0006}$ | $\mathbf{0.6625 \pm 0.0036}$ | $43.94 \pm 3.78$ |
| | 80% flips | NoTrans | $56.57 \pm 0.64$ | $0.9057 \pm 0.0033$ | $0.5622 \pm 0.0055$ | $33.60 \pm 3.04$ |
| | | LinearProbe | $19.46 \pm 0.75$ | $0.6895 \pm 0.0011$ | $0.1177 \pm 0.0108$ | $0.00 \pm 0.00$ |
| | | Adapter | $18.49 \pm 0.46$ | $0.6906 \pm 0.0006$ | $0.1219 \pm 0.0156$ | $0.00 \pm 0.00$ |
| | | Distill | $53.51 \pm 0.79$ | $0.8982 \pm 0.0021$ | $0.5269 \pm 0.0091$ | $26.80 \pm 2.49$ |
| | | LoRA | $22.92 \pm 1.73$ | $0.7202 \pm 0.0079$ | $0.1911 \pm 0.0308$ | $0.76 \pm 1.52$ |
| | | DANN-Gate | $20.83 \pm 1.32$ | $0.7097 \pm 0.0084$ | $0.1341 \pm 0.0253$ | $0.00 \pm 0.00$ |
| | | REFINE | $\mathbf{56.58 \pm 0.33}$ | $\mathbf{0.9067 \pm 0.0019}$ | $\mathbf{0.5655 \pm 0.0041}$ | $\mathbf{36.90 \pm 2.94}$ |
| | Schematic confusion | NoTrans | $56.53 \pm 0.77$ | $0.9006 \pm 0.0021$ | $0.5639 \pm 0.0056$ | $35.76 \pm 2.75$ |
| | | LinearProbe | $48.54 \pm 0.42$ | $0.8987 \pm 0.0008$ | $0.4757 \pm 0.0046$ | $18.44 \pm 7.89$ |
| | | Adapter | $47.17 \pm 0.82$ | $0.8998 \pm 0.0006$ | $0.4479 \pm 0.0148$ | $7.42 \pm 6.47$ |
| | | Distill | $57.80 \pm 0.44$ | $\mathbf{0.9068 \pm 0.0009}$ | $0.5772 \pm 0.0037$ | $35.92 \pm 3.00$ |
| | | LoRA | $49.96 \pm 0.26$ | $0.9039 \pm 0.0005$ | $0.4864 \pm 0.0116$ | $16.34 \pm 9.91$ |
| | | DANN-Gate | $49.04 \pm 0.33$ | $0.9028 \pm 0.0006$ | $0.4719 \pm 0.0059$ | $11.40 \pm 1.53$ |
| | | REFINE | $\mathbf{58.65 \pm 0.47}$ | $0.9034 \pm 0.0011$ | $\mathbf{0.5861 \pm 0.0048}$ | $\mathbf{38.40 \pm 3.10}$ |
| | Class imbalance | NoTrans | $56.44 \pm 0.48$ | $0.9055 \pm 0.0019$ | $0.5599 \pm 0.0051$ | $32.80 \pm 4.54$ |
| | | LinearProbe | $53.15 \pm 1.04$ | $0.8883 \pm 0.0145$ | $0.5238 \pm 0.0215$ | $28.36 \pm 14.04$ |
| | | Adapter | $51.64 \pm 0.99$ | $0.8960 \pm 0.0022$ | $0.5130 \pm 0.0150$ | $19.52 \pm 8.32$ |
| | | Distill | $54.89 \pm 0.49$ | $0.9063 \pm 0.0013$ | $0.5492 \pm 0.0065$ | $\mathbf{41.96 \pm 3.43}$ |
| | | LoRA | $53.21 \pm 0.19$ | $0.8975 \pm 0.0005$ | $0.5338 \pm 0.0022$ | $33.76 \pm 5.38$ |
| | | DANN-Gate | $53.05 \pm 0.28$ | $0.8964 \pm 0.0009$ | $0.5281 \pm 0.0055$ | $32.62 \pm 3.60$ |
| | | REFINE | $\mathbf{56.54 \pm 0.73}$ | $\mathbf{0.9103 \pm 0.0012}$ | $\mathbf{0.5619 \pm 0.0103}$ | $31.58 \pm 10.31$ |

Table 2: Single-source transfer learning with label noise, semantic perturbation, and class imbalance for CIFAR-10 using CNNs.

in the pretraining data while keeping the target domain fixed: (i) heavy label noise with 40% random label flips, (ii) extreme label noise with 80% flips, (iii) semantic perturbation created by paired-class flipping combined with additive image noise, and (iv) class imbalance induced by resampling to a long-tailed distribution. In addition, we repeat the experiments on CIFAR-100 and also evaluate both CIFAR-10 and CIFAR-100 with transformer-based models. We report the corresponding results in Appendix C.3.

Table 2 summarizes the results. REFINE consistently mitigates severe degradation and outperforms competing methods across all stress-test scenarios. In the moderate noise setting with 40% label flips, it achieves the best overall balance, improving accuracy and F1 by about 1% over Adapter and LoRA, while maintaining competitive minimum class accuracy. In the more extreme noise setting with 80% flips, most baselines collapse, with LinearProbe, Adapter, and DANN-Gate drop below 25% accuracy, whereas REFINE remains close to the no-transfer baseline, improving accuracy by nearly 35% over the strongest adaptive alternative. In the semantic confusion setting, with paired-class flips plus image noise, REFINE gains 1-2% in accuracy and F1 over NoTrans, while all other adaptive baselines perform worse, highlighting the robustness of REFINE to perturbed label semantics. In the class imbalance setting, it surpasses LinearProbe, Adapter, and LoRA by 3-5% in accuracy and F1, achieving the strongest overall results aside from a slightly lower minimum class accuracy than Distillation. Overall, REFINE avoids the catastrophic failures common to existing transfer strategies under noise, semantic perturbation, and class imbalance, while consistently delivering performance gains across all stress-test conditions.

We also briefly remark that, an important advantage of REFINE is that its complexity can be flexibly tuned through the choice of the encoder $h$. Such a design keeps it comparable in parameter efficiency to methods such as Adapter and Distillation. For instance, in this setting, for REFINE, the number of trainable parameters is $4.88\%$ of the total number of parameters in the frozen source model, for Adapter, it is $5.46\%$, and for Distillation, $4.68\%$. Thus REFINE achieves a comparable parameter efficiency, but clearly outperforms in mitigating negative transfer. The ablation study in Appendix C.5 further shows that the performance of REFINE remains stable across different parameter choices of $h$, indicating that the overall parameter complexity has relatively little impact.

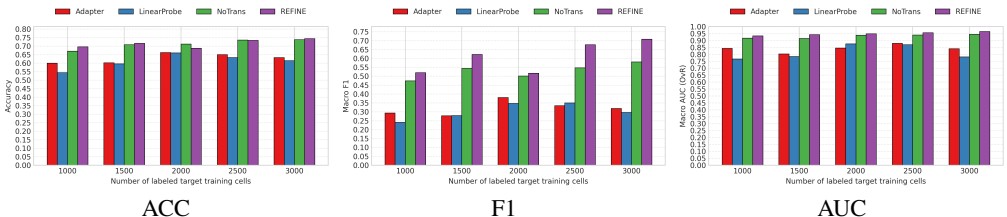

Figure 2: Metric comparison across labeled target sizes for Adapter, LinearProbe, NoTrans, and REFINE.

By contrast, increasing Adapter's complexity fails to resolve negative transfer, suggesting that its limitation stems from design rather than capacity.

## 5.4 ADAPT-TIME MULTI-MODALITY EXTENSION: SPATIAL-OMICS EXAMPLE

Spatial transcriptomics provides a clean setting to study a phenomenon that is becoming increasingly common in biological data analysis: a pretrained model is strong, but a crucial modality appears only at adaptation time. In the SpatialGlue human lymph node dataset [30], cells come with both transcriptomes and spatial coordinates, and the major anatomical domains cortex, medulla cords, follicles, capsule, pericapsular adipose tissue, and others form well structured spatial patterns. Because expert annotation of these domains is costly, only a small subset of cells typically receives labels. Importantly, scGPT [7], like most foundation models for single cell data, is pretrained purely on dissociated RNA and therefore never observes spatial information. This naturally creates what we refer to as an adapt-time multimodality extension problem, where the model must incorporate a modality that was entirely absent during pretraining. Conventional fine-tuning or PEFT cannot reliably supply information the pretrained model has never learned.

The empirical results reflect this challenge. As shown in Figure 2, both LinearProbe and Adapter exhibit negative transfer when applied directly to scGPT representations. With 1000 labeled cells, their F1 scores remain near 0.24 to 0.29, far below a simple GNN trained from scratch, denoted as NoTrans, which already reaches about 0.47 by leveraging spatial structure directly. Even with more labeled data, their AUC and F1 improvements stagnate. In contrast, REFINE adds a lightweight residual spatial encoder that complements the frozen scGPT features without modifying the backbone. This allows the model to integrate the missing spatial modality at adaptation time. The gains are substantial: at 1000 labels, REFINE raises F1 to roughly 0.52, and surpasses 0.70 by 3000 labels, with consistently stronger AUC across all training sizes.

Figure S2 illustrates the qualitative impact of this adapt-time modality gap. LinearProbe and Adapter compress the cortical region, miss several follicular and trabeculae regions. REFINE reconstructs these domains much more faithfully, recovering cortical extent, follicle structure, and peripheral regions that the other approaches systematically miss. These results suggest a broader message. When a modality is entirely missing during pretraining, fine-tuning alone is insufficient, but a residual mechanism that injects the missing information at adaptation time can bridge the gap effectively and without imposing additional engineering burdens on practitioners.

## ETHICS STATEMENT

This research does not involve human subjects, personally identifiable information, or sensitive data. The datasets used are publicly available and widely used in the community. We are not aware of direct applications of our method that raise ethical concerns. Nevertheless, as with any machine learning system, there is a potential risk of misuse if deployed in contexts where fairness or bias are critical. We encourage future work to examine these dimensions before deployment in such settings.

REPRODUCIBILITY STATEMENT

We have made efforts to ensure the reproducibility of our results. Detailed descriptions of datasets, preprocessing steps, and hyperparameters, optimizers are provided in Section 5 and Appendix D. All proofs for theoretical claims are provided in Section 4 and Appendix A. An anonymized version of our source code is included in the supplementary materials and will be released publicly upon acceptance.

ACKNOWLEDGMENT

The research of LZ was partially supported by NSF CAREER DMS-2340241 and Renaissance Philanthropy "AI for Math" Fund. The research of LL was partially supported by NIH grants UG3NS140730 and R01AG080043.

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

APPENDICES

In the appendices, we provide additional technical and empirical details. Appendix A provides the proof of the main theorem, supported by auxiliary lemmas in Appendix B. Appendix C expands the empirical evaluations, including additional results on more benchmark data, tabular data, and an ablation study. Appendix D documents the experiment setup and implementation details for reproducibility. Together, they offer a complete account of the theory, validation, and practical details underlying our work.

## A PROOFS OF MAIN RESULTS

In this section, we prove the main results in Section 4.

**Additional Notation.** Let $\|\cdot\|_{L_q}$ denote the $L_q$ norm under the probability measure $\mathbb{P}_X^t$ for any $q \in [1,\infty]$, where $\mathbb{P}_X^t$ is the distribution of $X_i$ in the training data. For $a, b \in \mathbb{R}$, we define $a \wedge b = \min\{a, b\}$ and $a \vee b = \max\{a, b\}$.

In addition, we would like to recall $\mathcal{R}_{\mathbb{P}^t}(g) = \mathbb{E}_{(X,Y)\sim\mathbb{P}^t}[(g(X) - Y)^2]$. As a result,

$$
\begin{aligned}
\mathcal{R}_{\mathbb{P}^t}(g) - \mathcal{R}_{\mathbb{P}^t}(f^*) &= \mathbb{E}_{(X,Y)\sim\mathbb{P}^t}[(g(X) - f^*(X) - \epsilon)^2] - \sigma^2 \\
&= \mathbb{E}_{(X,Y)\sim\mathbb{P}^t}[(g(X) - f^*(X))^2] \asymp \|g - f^*\|_{L_2}^2,
\end{aligned}
\tag{S.1}
$$

where the last equivalence $\asymp$ is due to the fact that we assume $X$ has positive continuous density on $[0,1]^d$ bounded by an absolute value. As $[0,1]^d$ is a compact space and the density function of $X$ is continuous, this implies that the density function is both upper and lower bounded by absolute constants.

### A.1 GENERALIZATION ERROR UPPER BOUND

We now prove the main theorem on the prediction risk of REFINE. The results of the two corollaries can be obtained straightforwardly, and we thus omit their proofs.

*Proof of Theorem 4.1.* Recall that $v^*$ is defined as the optimal linear probe of $f_{\text{rep}}$, i.e.,

$$
v^* = \arg\min_{v \in \mathbb{R}^p} \mathbb{E}[\{f^*(X) - v^\top f_{\text{rep}}(X)\}^2].
$$

We begin by observing that the difficulty of the estimation problem is governed by the residual $r^* := f^* - v^{*\top} f_{\text{rep}}$, since $f_{\text{rep}}$ is assumed to be known, and $v^{*\top} f_{\text{rep}}$ can be seen as a linear function of the known quantity. By appropriately choosing the parameters $W$, $L$, and $B$, we control the complexity of the neural network, and the bias of estimating $r^*$.

Specifically, choose

$$
L = (2 + \lceil \log_2 \beta \rceil)\left(11 + \frac{\beta}{d}\right), \quad W = c_1' \epsilon^{-d/\beta}, \quad B = (\rho^* \vee 1)\epsilon^{-c_2'},
\tag{S.2}
$$

where $c_1', c_2' > 0$ are constants appearing in Lemma B.2. Define $\rho^* := \|r^*\|_{\mathcal{C}^\beta}$. Set

$$
\epsilon := n^{-\beta/(2\beta+d)} \rho^{-2\beta/(2\beta+d)} \wedge 1,
\tag{S.3}
$$

where $\rho > 0$ is some tuning parameter. The choices in (4) are realized by taking $\epsilon = n^{-\beta/(2\beta+d)} \rho^{-2\beta/(2\beta+d)}$ and setting $c_1 := (2 + \lceil \log_2 \beta \rceil)(11 + \beta/d)$, $c_2 := c_1'$, $c_3 := c_2'$.

Note that $\sup_{g\in\mathcal{G}_{d,p}(W,L,B;f_{\text{rep}})} \|g\|_{L_\infty} \leq 2$. From Lemma B.3 with the choice $\delta \leftarrow 1/n$, we have

$$
\mathbb{E}[\|\hat{g} - f^*\|_{L_2}^2] \lesssim \left(\inf_{g\in\mathcal{G}} \|g - f^*\|_{L_2}^2 + \frac{\log \mathcal{N}(1/n, \mathcal{G}_{d,p}(W, L, B; f_{\text{rep}}), \|\cdot\|_{L_\infty})}{n} + \frac{1}{n}\right).
$$

Next we compute the first term and the second term separately.

**Part 1: Bounding the first term.** Suppose for now that $\rho^* > 0$. Rescale the residual by noting that $(1/\rho^*)r^* \in \mathcal{C}_u^\beta$. Then, by Lemma B.2, there exists a neural network $r_{\mathrm{NN}} \in \mathcal{H}_d(W, L, B)$ such that

$$\|r_{\mathrm{NN}} - (1/\rho^*)r^*\|_{L_2} \lesssim \epsilon. \tag{S.4}$$

This inequality provides the approximation error of the ReLU network class. To translate this result to the bias term $\|g - f^*\|_{L_2}^2$, we proceed as follows. Write

$$r_{\mathrm{NN}} = r_{\mathrm{NN},L} \circ r_{\mathrm{NN},L-1} \circ \cdots \circ r_{\mathrm{NN},1}(x),$$

where $r_{\mathrm{NN},\ell}(x) = \sigma(A_\ell x + b_\ell)$ for $\ell \in [L-1]$ and $r_{\mathrm{NN},L}(x) = A_L x + b_L$. Define $r'_{\mathrm{NN},L}(x) = (\rho^* A_L)x + (\rho^* b_L)$ to approximate $\rho^* r_{\mathrm{NN}}$. Then, it follows that the function

$$g^\circ(x) := 1 \wedge ((-1) \vee r'_{\mathrm{NN},L} \circ r_{\mathrm{NN},L-1} \circ \cdots \circ r_{\mathrm{NN},1}(x)) + v^{*\top} f_{\mathrm{rep}}(x)$$

belongs to $\mathcal{G}_{d,p}(W, L, B; f_{\mathrm{rep}})$ since $\|v^*\| \leq 1$. Moreover, we can write $g^\circ$ as

$$g^\circ(x) = 1 \wedge ((-1) \vee \rho^* r_{\mathrm{NN}}(x)) + v^{*\top} f_{\mathrm{rep}}(x).$$

Using (S.4), we have

$$\mathbb{E}[\{g^\circ(X_1) - f^*(X_1)\}^2]^{1/2} = \|1 \wedge ((-1) \vee \rho^* r_{\mathrm{NN}}) + v^{*\top} f_{\mathrm{rep}} - f^*\|_{L_2}$$

$$= \rho^* \left\| \frac{1}{\rho^*} \wedge \left( -\frac{1}{\rho^*} \vee r_{\mathrm{NN}} \right) - \frac{1}{\rho^*} r^* \right\|_{L_2}$$

$$\leq \rho^* \left\| r_{\mathrm{NN}} - \frac{1}{\rho^*} r^* \right\|_{L_2}$$

$$\lesssim \rho^* \epsilon,$$

where we used the fact that $\|r^*/\rho^*\|_{L_\infty} \leq \|r^*/\rho^*\|_{\mathcal{C}^\beta} \leq 1/\rho^*$. Thus,

$$\inf_{g \in \mathcal{G}_{d,p}(W,L,B;f_{\mathrm{rep}})} \mathbb{E}[\|g - f^*\|_{L_2}^2] \leq \mathbb{E}[\|g^\circ - f^*\|_{L_2}^2] \lesssim \rho^{*2} \epsilon^2. \tag{S.5}$$

If instead $\rho^* = 0$, then we can simply choose a ReLU network in $\mathcal{H}_d(W, L, B)$ with all weights and biases set to zero. By taking $g^\circ = 0 + v^{*\top} f_{\mathrm{rep}}$, the bound in equation S.5 trivially holds.

**Part 2: Bounding the second term.** The covering number bound from Lemma B.4 with the choice of $W, L, B$ in (S.2), we have

$$\frac{\log \mathcal{N}(1/n, \mathcal{G}_{d,p}(W, L, B; f_{\mathrm{rep}}), \|\cdot\|_{L_\infty})}{n} \leq \frac{C'}{n}(\epsilon^{-d/\beta} + p) \log\left(\frac{n}{\epsilon}\right),$$

where $C'$ is a constant depending on $d$ and $\beta$.

**Part 3: Balancing terms.** Finally, we combine the results from part 1 and part 2. Recalling the choice of $\epsilon$ in (S.3), we consider two cases depending on the value of $\rho$.

When $1/\sqrt{n} \leq \rho$, we have $\epsilon = (n\rho^2)^{-\beta/(2\beta+d)}$. In this case, the bound becomes

$$\mathbb{E}[\|\hat{g} - f^*\|_{L_2}^2] \leq \rho^{*2} \rho^{-4\beta/(2\beta+d)} n^{-2\beta/(2\beta+d)} + C'\left(\rho^{2d/(2\beta+d)} n^{-2\beta/(2\beta+d)} + \frac{p}{n}\right) \log n$$

$$\leq (C'+1)\left((\rho^{*2}\rho^{-4\beta/(2\beta+d)} + \rho^{2d/(2\beta+d)} \log n)n^{-2\beta/(2\beta+d)} + \frac{p \log n}{n}\right). \tag{S.6}$$

When $\rho \leq 1/\sqrt{n}$ (so that $\epsilon = 1$), the bound becomes

$$\mathbb{E}[\|\hat{g} - f^*\|_{L_2}^2] \leq \rho^{*2} + C'\frac{p \log n}{n} \leq (C'+1)\left(\rho^{*2}\rho^{-4\beta/(2\beta+d)} n^{-2\beta/(2\beta+d)} + \frac{p \log n}{n}\right). \tag{S.7}$$

Combining the bounds in (S.6) and (S.7) with (S.1), we obtain the desired result.

This completes the proof of Theorem 4.1. □

## A.2 WORST CASE NO-NEGATIVE-TRANSFER GUARANTEE

*Proof of Corollary 4.4.* By a similar argument as in the proof of Theorem 4.1, given any $f_{\text{rep}}$ satisfying $v^\top f_{\text{rep}} \in \mathcal{C}_u^\beta$ for all unit vector $v \in \mathbb{S}^{p-1}$, the prediction risk can be upper bounded as

$$\sup_{f^* \in \mathcal{F}^\beta(f_{\text{rep}}, \gamma)} \mathbb{E}[\mathcal{R}_{\mathbb{P}^t}(\hat{g}) - \mathcal{R}_{\mathbb{P}^t}(f^*)] \lesssim \gamma^{\frac{2d}{2\beta+d}} n^{-\frac{2\beta}{2\beta+d}} \log n + \frac{p \log n}{n}, \tag{S.8}$$

For the linear adapter, for any estimator $\hat{w}$, the prediction risk with respect to $f^*$ satisfies

$$\mathbb{E}\big[(\hat{w}^\top X_1 - f^*(X_1))^2\big] = \mathbb{E}\big[(w^{*\top} X_1 - f^*(X_1))^2\big] + \mathbb{E}\big[(\hat{w} - w^*)^\top \mathbb{E}[X_1 X_1^\top](\hat{w} - w^*)\big],$$

where $w^* = \arg\min_{w \in \mathbb{R}^p} \mathbb{E}[(w^\top X - f^*(X))^2]$. This follows by the normal equation

$$\mathbb{E}\big[X(f^*(X) - w^{*\top} X)\big] = 0,$$

which imply that the cross term vanishes. Under the model in equation 1, the estimation term is of order $\Theta(p/n)$. Hence we have

$$\sup_{f^* \in \mathcal{F}^\beta(f_{\text{rep}}, \gamma)} \mathbb{E}[\mathcal{R}_{\mathbb{P}^t}(\hat{w}_{\text{ft}}^\top f_{\text{rep}}) - \mathcal{R}_{\mathbb{P}^t}(f^*)] \gtrsim \sup_{f^* \in \mathcal{F}^\beta(f_{\text{rep}}, \gamma)} \inf_{v \in \mathbb{R}^p} \mathbb{E}[\|v^\top f_{\text{rep}}(X_1) - f^*(X_1)\|^2] + \frac{p}{n}. \tag{S.9}$$

For the training from scratch, note that $\gamma \mathcal{C}_u^\beta \subset \mathcal{F}^\beta(f_{\text{rep}}, \gamma)$. By Theorem 3.2 in Györfi et al. [13], we have

$$\sup_{f^* \in \mathcal{F}^\beta(f_{\text{rep}}, \gamma)} \mathbb{E}[\mathcal{R}_{\mathbb{P}^t}(\hat{g}_{\text{sc}}) - \mathcal{R}_{\mathbb{P}^t}(f^*)] \geq \inf_{\check{g}} \sup_{r^* \in \gamma \mathcal{C}_u^\beta} \mathbb{E}[(\check{g}(X_1) - r^*(X_1))^2] \gtrsim \gamma^{2d/(2\beta+d)} n^{-2\beta/(2\beta+d)}. \tag{S.10}$$

Therefore, combining equation S.9 and equation S.10 with equation S.8 concludes the proof. $\square$

## A.3 ASYMPTOTIC NO-NEGATIVE-TRANSFER GUARANTEE

Here we compare the performance of REFINE with two natural baselines: training from scratch with comparable model capacity, and fitting a linear probe on $f_{\text{rep}}$, without any parameter tuning.

**Proposition A.1** (Asymptotic No-negative-transfer Guarantee). *Assume $v^{*\top} f_{rep} - f^* \in \mathcal{C}^\beta$. Suppose that the parameters $(W, L, B)$ satisfy $W \log(nLB^L(W+1)^L) = o(n)$ and that $p \log n = o(n)$ as $n \to \infty$. Consider the model trained from scratch and the linear probe on $f_{rep}$ with comparable capacity:*

$$\hat{g}_{sc} = \arg\min_{g \in \mathcal{H}_d(W, L, B)} \frac{1}{n} \sum_{i \in [n]} \ell(g(X_i), Y_i), \quad \hat{w}_{ft} = \arg\min_{w \in \mathbb{R}^p, \|w\| \leq 1} \frac{1}{n} \sum_{i \in [n]} \ell(w^\top f_{rep}(X_i), Y_i). \tag{S.11}$$

*Then,*

$$\mathbb{E}[\mathcal{R}_{\mathbb{P}^t}(\hat{g}) - \mathcal{R}_{\mathbb{P}^t}(f^*)] \leq (1 + o(1)) \min\{\mathbb{E}[\mathcal{R}_{\mathbb{P}^t}(\hat{g}_{sc}) - \mathcal{R}_{\mathbb{P}^t}(f^*)], \mathbb{E}[\mathcal{R}_{\mathbb{P}^t}(\hat{w}_{ft}^\top f_{rep}) - \mathcal{R}_{\mathbb{P}^t}(f^*)]\} + o(1), \tag{S.12}$$

*as $n \to \infty$.*

Proposition A.1 shows that, provided the model capacity increases slowly enough with $n$ so that the estimation error vanishes, the excess risk of REFINE is bounded, up to multiplicative and an additive $o(1)$ term, by the smaller of the two alternatives: training a comparable model from scratch or fitting a linear probe on $f_{\text{rep}}$. In particular, REFINE is asymptotically no worse than either baseline for *any* moderate choice of $(W, L, B)$. We also note that the choice of $(W, L, B)$ in the following theorem satisfies the conditions of Proposition A.1.

We now prove the proposition that REFINE avoids negative transfer asymptotically, which provides a result when either $\mathcal{R}_{\mathbb{P}^t}(\hat{w}_{\text{ft}}^\top f_{\text{rep}})$ or $\mathcal{R}_{\mathbb{P}^t}(\hat{g}_{\text{sc}})$ is bounded from below as $n \to \infty$.

*Proof of Proposition A.1.* Consider the hypothesis classes for training from scratch and linear probing:

$$\mathcal{G}_{\text{sc}}(W, L, B) := \{x \mapsto h(x) : h \in \bar{\mathcal{H}}_d(W, L, B)\}, \qquad \mathcal{G}_{\text{ft}} := \{x \mapsto v^\top f_{\text{rep}}(x) : \|v\| \leq 1\}.$$

Let $\hat{g}_{\text{sc}}$ and $\hat{g}_{\text{ft}}$ be the empirical risk minimizers over $\mathcal{G}_{\text{sc}}(W, L, B)$ and $\mathcal{G}_{\text{ft}}$, respectively. To ease notation, we write $\mathcal{G} = \mathcal{G}_{d,p}(W, L, B; f_{\text{rep}})$. By construction, we have $\mathcal{G}_{\text{ft}} \cup \mathcal{G}_{\text{sc}} \subset \mathcal{G}$. Hence

$$\inf_{g \in \mathcal{G}} \mathcal{R}_{\mathbb{P}^t}(g) \leq \min\{\inf_{g \in \mathcal{G}_{\text{sc}}} \mathcal{R}_{\mathbb{P}^t}(g), \inf_{g \in \mathcal{G}_{\text{ft}}} \mathcal{R}_{\mathbb{P}^t}(g)\} \leq \min\{\mathbb{E}[\mathcal{R}_{\mathbb{P}^t}(\hat{g}_{\text{sc}})], \mathbb{E}[\mathcal{R}_{\mathbb{P}^t}(\hat{g}_{\text{ft}})]\}. \tag{S.13}$$

By assumption, $\|f^*\|_{L_\infty} \leq 1$ and $\sup_{g \in \mathcal{G}} \|g\|_{L_\infty} \leq 2$. Lemma B.3 with the choice $\delta \leftarrow 1/n$ and $F \leftarrow 4$ gives

$$\mathbb{E}\big[\mathcal{R}_{\mathbb{P}^t}(\hat{g}) - \mathcal{R}_{\mathbb{P}^t}(f^*)\big] \leq (1 + \kappa)^2 (\inf_{g \in \mathcal{G}} \mathcal{R}_{\mathbb{P}^t}(g) - \mathcal{R}_{\mathbb{P}^t}(f^*)) + C_1 \Big(\frac{\log \mathcal{N}(1/n, \mathcal{G}, \|\cdot\|_{L_\infty})}{n\kappa} + \frac{1}{n}\Big), \tag{S.14}$$

for some universal constant $C_1 > 0$, where we used $\mathcal{R}_{\mathbb{P}^t}(g) - \mathcal{R}_{\mathbb{P}^t}(f^*) = \mathbb{E}[(g(X_1) - f^*(X_1))^2]$.

Lemma B.4 with the choice $\delta \leftarrow 1/n$ gives

$$\log \mathcal{N}(1/n, \mathcal{G}, \|\cdot\|_{L_\infty}) \leq C_2 \Big\{ W \log \Big(nLB^L(W + 1)^L\Big) + p \log n \Big\}, \tag{S.15}$$

for some universal constant $C_2 > 0$. Combining equation S.13 and equation S.15 into the right hand side of equation S.14 yields

$$\mathbb{E}\big[\mathcal{R}_{\mathbb{P}^t}(\hat{g}) - \mathcal{R}_{\mathbb{P}^t}(f^*)\big] \leq (1 + \kappa)^2 \min\{\mathbb{E}[\mathcal{R}_{\mathbb{P}^t}(\hat{g}_{\text{sc}})] - \mathcal{R}_{\mathbb{P}^t}(f^*), \mathbb{E}[\mathcal{R}_{\mathbb{P}^t}(\hat{g}_{\text{ft}})] - \mathcal{R}_{\mathbb{P}^t}(f^*)\}$$

$$+ C\left\{ \frac{1}{\kappa} \left( \frac{W \log\big(nLB^L(W + 1)^L\big)}{n} + \frac{p \log n}{n} \right) + \frac{1}{n} \right\},$$

where $C > 0$ is some universal constant. Since $W \log\big(nLB^L(W + 1)^L\big) = o(n)$ and that $p \log n = o(n)$ as $n \to \infty$, we have

$$\mathbb{E}\big[\mathcal{R}_{\mathbb{P}^t}(\hat{g}) - \mathcal{R}_{\mathbb{P}^t}(f^*)\big] \leq (1 + \kappa)^2 \min\{\mathbb{E}[\mathcal{R}_{\mathbb{P}^t}(\hat{g}_{\text{sc}})] - \mathcal{R}_{\mathbb{P}^t}(f^*), \mathbb{E}[\mathcal{R}_{\mathbb{P}^t}(\hat{g}_{\text{ft}})] - \mathcal{R}_{\mathbb{P}^t}(f^*)\} + \frac{R}{\kappa},$$

where $R = o(1)$ as $n \to \infty$. Since $\kappa \in (0, 1]$ is arbitrary, we choose $\kappa = \sqrt{R} \wedge 1(= o(1))$ to conclude the proof. □

## B   AUXILIARY LEMMAS

In this section, we provide some auxiliary lemmas.

The next lemma is about the entropy bound for $\mathcal{H}_d(W, L, B)$.

**Lemma B.1** (Lemma 21 from Nakada & Imaizumi [33]). *Fix any W, L, and B > 0. Then, we have the covering number bound*

$$\log \mathcal{N}(\epsilon, \mathcal{H}_d(W, L, B), \|\cdot\|_{L_\infty}) \leq W \log \left( \frac{2LB^L(W + 1)^L}{\epsilon} \right).$$

The next lemma is modified from Petersen & Voigtlaender [37], adapted to consider $L_2$ approximation error with respect to the probability measure $\mathbb{P}_X^t$ over the domain $[0, 1]^d$, rather than the original $L_2$ error with a uniform measure on $[-1/2, 1/2]^d$.

**Lemma B.2** (Modification of Theorem 3.1 from Petersen & Voigtlaender [37]). *Fix $d \in \mathbb{N}_+$ and $\beta > 0$. Suppose that $\mathbb{P}_X^t$ has a density bounded by $O(1)$. Then, there exist constants $c_1', c_2' > 0$, depending on $d$ and $\beta$, such that for any $\epsilon \in (0, 1/2)$, if one chooses W, L, and B satisfying*

$$L \leq (2 + \lceil \log_2 \beta \rceil)\Big(11 + \frac{\beta}{d}\Big), \quad W \leq c_1' \epsilon^{-d/\beta}, \quad B \leq \epsilon^{-c_2'},$$

*then*

$$\sup_{f^\# \in \mathcal{C}_u^\beta} \inf_{f_{NN} \in \mathcal{H}_d(W, L, B)} \|f_{NN} - f^\#\|_{L_2} \lesssim \epsilon.$$

The next lemma provides a bound on the prediction risk of the empirical risk minimizer in terms of the covering number of the function class and the approximation error.

**Lemma B.3** (Modification to Lemma 4 from Schmidt-Hieber [39]). *Let $\mathcal{G}$ be a function class, and let $\hat{g}$ be the minimizer of the empirical risk $(1/n)\sum_{i\in[n]}\ell(\hat{g}(X_i), Y_i)$ over $\mathcal{G}$ under the data generating process introduced in Section 4. Suppose that $\{f^*\}\cup\mathcal{G}\subset\{[0,1]^d\to[-F,F]\}$ for some $F\geq 1$. Then there exists a universal constant $C_0 > 0$ such that*

$$\mathbb{E}[\|\hat{g}-f^*\|_{L_2}^2] \leq (1+\kappa)^2\left\{\inf_{g\in\mathcal{G}}\|g-f^*\|_{L_2}^2 + C_0\left(F^2\frac{\log\mathcal{N}(\delta,\mathcal{G},\|\cdot\|_{L_\infty})}{n\kappa}+\delta F\right)\right\}$$

*for all $\kappa,\delta\in(0,1]$.*

The next lemma provides a bound on the covering number of the REFINE class $\mathcal{G}_{d,p}(W, L, B; f_{\text{rep}})$.

**Lemma B.4.** *Fix $W\in\mathbb{N}_+$, $L\in\mathbb{N}_+$, $B > 0$, and $\delta > 0$. Then, there exists a universal constant $C > 0$ such that*

$$\log\mathcal{N}(\delta,\mathcal{G}_{d,p}(W,L,B;f_{rep}),\|\cdot\|_{L_\infty}) \leq C\left\{W\log\left(\frac{LB^L(W+1)^L}{\delta}\right)+p\log\left(\frac{1}{\delta}\right)\right\}.$$

*Proof.* We next bound the covering number $\mathcal{N}(\delta,\mathcal{G}_{d,p}(W,L,B;f_{\text{rep}}),\|\cdot\|_{L_\infty})$. Note that for any $\delta > 0$, we have

$$\log\mathcal{N}(\delta,\mathcal{G}_{d,p}(W,L,B;f_{\text{rep}}),\|\cdot\|_{L_\infty})$$
$$\leq \log\mathcal{N}\left(\frac{\delta}{2},\{x\mapsto uh(x)\mid u\in[-1,1],h\in\bar{\mathcal{H}}_d(W,L,B)\},\|\cdot\|_{L_\infty}\right)$$
$$+\log\mathcal{N}\left(\frac{\delta}{2},\{x\mapsto v^\top f_{\text{rep}}(x)\mid v\in\mathcal{B}_p(1)\},\|\cdot\|_{L_\infty}\right). \tag{S.16}$$

Recall that $f_{\text{rep}}: [0,1]^d\to\mathcal{B}_p(1)$. Since $\|v^\top f_{\text{rep}}-v'^\top f_{\text{rep}}\|_{L_\infty}\leq\|v-v'\|$ for any $v,v'\in\mathcal{B}_p(1)$, a standard argument shows that

$$\mathcal{N}\left(\frac{\delta}{2},\{x\mapsto v^\top f_{\text{rep}}(x)\mid v\in\mathcal{B}_p(1)\},\|\cdot\|_{L_\infty}\right)\leq\mathcal{N}\left(\frac{\delta}{2},\mathcal{B}_p(1),\|\cdot\|\right)\leq\left(\frac{6}{\delta}\right)^p. \tag{S.17}$$

Furthermore, since $\|u_1 h_1 - u_2 h_2\|_{L_\infty}\leq\|h_1-h_2\|_{L_\infty}+|u_1-u_2|$ for any $u_1,u_2\in[-1,1]$ and $h_1,h_2\in\bar{\mathcal{H}}_d(W,L,B)$, we have

$$\mathcal{N}(\frac{\delta}{2},\{x\mapsto uh(x)\mid u\in[-1,1],h\in\bar{\mathcal{H}}_d(W,L,B)\},\|\cdot\|_{L_\infty})$$
$$\leq\mathcal{N}\left(\frac{\delta}{4},[-1,1],|\cdot|\right)\mathcal{N}\left(\frac{\delta}{4},\bar{\mathcal{H}}_d(W,L,B),\|\cdot\|_{L_\infty}\right)$$
$$\lesssim\frac{1}{\delta}\mathcal{N}\left(\frac{\delta}{4},\mathcal{H}_d(W,L,B),\|\cdot\|_{L_\infty}\right). \tag{S.18}$$

Note that clipping does not increase the covering number of $m H_d(W, L, B)$. Using (S.16), (S.17) and (S.18), combined with Lemma B.1, we obtain

$$\log\mathcal{N}(\delta,\mathcal{G}_{d,p}(W,L,B;f_{\text{rep}}),\|\cdot\|_{L_\infty})\lesssim W\log\left(\frac{LB^L(W+1)^L}{\delta}\right)+p\log\left(\frac{1}{\delta}\right).$$

This completes the proof of Lemma B.4. $\qquad\square$

## C  MORE NUMERICAL EXPERIMENTS

In this section, we present additional results that complement Section 5.

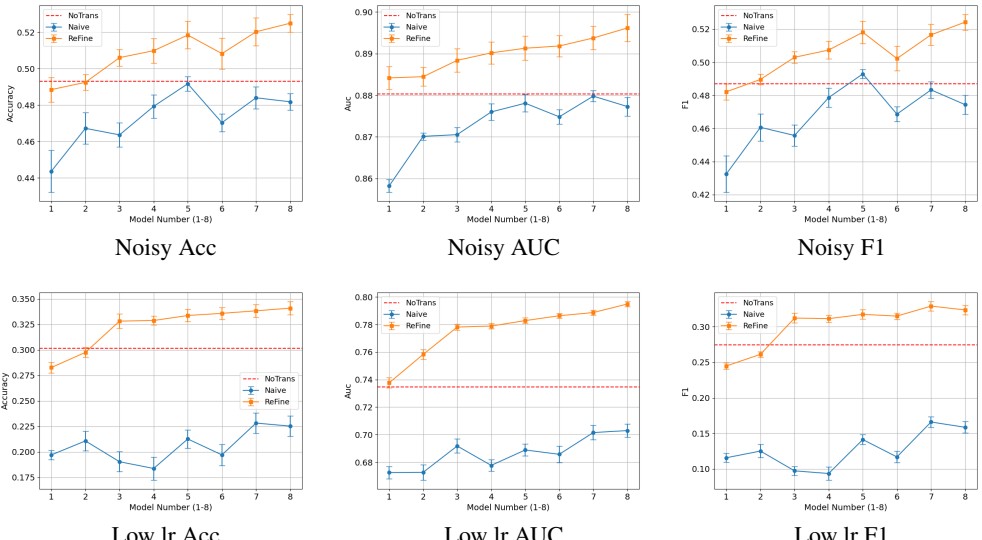

Figure S1: Results of multi-source transfer learning under noisy and low-learning-rate conditions.

## C.1 MULTI-SOURCE TRANSFER

In the third experiment setting, we investigate multi-source transfer, an important yet underexplored setting where knowledge is drawn from multiple heterogeneous sources to achieve better generalization than any single source alone. Despite its practical relevance, most existing approaches, such as LinearProbe, Adapter, and Distillation, are designed for single-source transfer and do not naturally extend to the multi-source case. To provide a fair comparison, we implement a Naive baseline that assigns each source domain its own feature extractor, concatenates the resulting representations, and trains a classifier on top of the joint embedding. This straightforward strategy captures the most natural way of leveraging multiple sources in the absence of specialized methods. For our experiments, we partition CIFAR-10 into eight disjoint subsets of 2000 samples each, treating them as distinct source domains and training separate CNNs on each. REFINE then integrates the corresponding penultimate representations through its modular structure, mimicking multi-source transfer while keeping inference overhead modest. This setup enables a direct evaluation of principled multi-source integration against naive concatenation.

Figure S1 reports the results under two stress conditions, a noisy case with $50\%$ label corruption, testing robustness to unreliable label supervision, and a low learning rate case, testing training stability and efficiency. In the noisy case, REFINE significantly outperforms both Naive and NoTrans as more external sources are integrated. With all eight sources, REFINE achieves classification accuracy $52.5\%$, AUC $0.8962$, and F1 $0.5242$, compared to Naive's $48.2\%$, $0.8773$, and $0.4744$, and NoTrans's $49.3\%$, $0.8803$, and $0.4871$. Notably, Naive consistently performs worse than NoTrans, indicating negative transfer when external information is not integrated effectively. In the low learning rate case, REFINE again improves steadily over NoTrans as the number of sources increases, while Naive suffers severe degradation. With all eight sources, REFINE reaches $34.09\%$ classification accuracy, surpassing NoTrans's $30.16\%$ and Naive's $22.53\%$. Overall, these results demonstrate that REFINE effectively integrates multiple sources, and remains robust under adverse supervision and training conditions. It avoids the pitfalls of naive concatenation and provides a stable approach for multi-source transfer.

## C.2 DISCUSSION ABOUT MULTIMODALITY EXTENSION

Baltrusaitis et al. [2] survey multimodal machine learning from a general taxonomy perspective, organizing existing methods into representation, translation, alignment, fusion, and co-learning paradigms. Gao et al. [8] focus specifically on deep multimodal learning techniques that emphasize neural joint representation learning and fusion strategies, assuming that all participating modalities

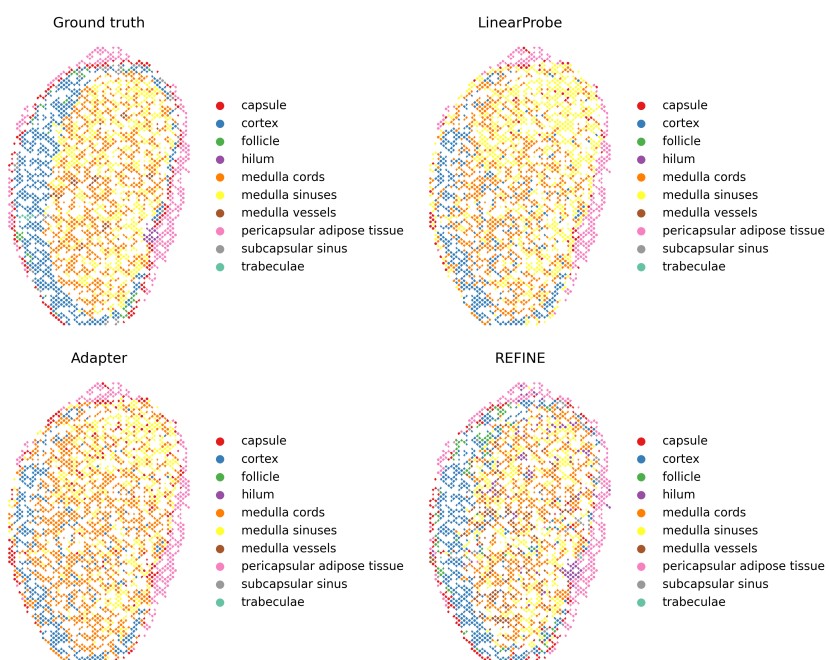

Figure S2: Spatial predictions at target cell.

are known and available during training. Stahlschmidt et al. [41] review biomedical multimodal fusion approaches that similarly rely on paired multimodal data and end-to-end or coordinated training with all modalities present beforehand. In contrast to these settings, we define an *adapt-time multimodality extension* regime in which a foundation model is pretrained on a single modality, the backbone remains fixed, upstream data are inaccessible, and a previously unseen modality becomes available only at adaptation time; to our knowledge, this problem formulation is not explicitly identified or studied in prior multi-source transfer or multimodal learning literature.

Figure S2 provides additional qualitative comparisons of spatial domain predictions on the human lymph node dataset, showing ground truth alongside results from LinearProbe, Adapter, and RE-FINE, as discussed in Section 5.4.

### C.3 SINGLE-SOURCE TRANSFER UNDER CHALLENGING SCENARIOS

Similar to the setting considered in Section 5.3 for CIFAR-10, we run the experiments on CIFAR-100. Moreover, in addition to CNNs, we also evaluate both CIFAR-10 and CIFAR-100 with transformer-based models.

Table S1 reports the results on CIFAR-100 with CNNs. Similar to CIFAR-10, REFINE consistently outperforms the baseline methods under all four stress scenarios. In particular, in the extreme noise setting with $80\%$ label flips, most competing methods collapse to near-random performance, whereas REFINE remains stable and comparable to the no-transfer baseline. In the semantic confusion and class imbalance settings, REFINE achieves the strongest improvements in classification accuracy and F1, highlighting its ability to mitigate negative transfer even when pretraining data is severely perturbed.

Table S2 and Table S3 report the results on CIFAR-10 and CIFAR-100, respectively, with transformer-based models. Similar to CNNs, existing adaptation methods degrade sharply under noisy or imbalanced pretraining, whereas REFINE maintains stable and superior performance in accuracy, AUC, and F1.

Together, these results demonstrate that the advantages of REFINE are not tied to a specific model architecture or dataset size. By design, it reliably suppresses negative transfer and delivers consistent gains under challenging pretraining conditions.

| Dataset | Setting | Method | Acc | AUC | F1 | MinCAcc |
|---------|---------|--------|-----|-----|-----|---------|
| CIFAR-100 | 40% flips | NoTrans | $17.82 \pm 0.36$ | $0.8259 \pm 0.0068$ | $0.1684 \pm 0.0039$ | $\mathbf{0.60 \pm 0.49}$ |
| | | LinearProbe | $17.35 \pm 0.27$ | $\mathbf{0.8605 \pm 0.0015}$ | $0.1472 \pm 0.0043$ | $0.00 \pm 0.00$ |
| | | Adapter | $16.19 \pm 0.33$ | $0.8578 \pm 0.0019$ | $0.1303 \pm 0.0037$ | $0.00 \pm 0.00$ |
| | | Distill | $18.73 \pm 0.22$ | $\mathbf{0.8605 \pm 0.0035}$ | $0.1631 \pm 0.0024$ | $0.00 \pm 0.00$ |
| | | LoRA | $17.24 \pm 0.33$ | $0.8568 \pm 0.0018$ | $0.1463 \pm 0.0053$ | $0.00 \pm 0.00$ |
| | | DANN-Gate | $15.02 \pm 0.39$ | $0.8472 \pm 0.0020$ | $0.1239 \pm 0.0041$ | $0.00 \pm 0.00$ |
| | | REFINE | $\mathbf{19.28 \pm 0.34}$ | $0.8555 \pm 0.0042$ | $\mathbf{0.1805 \pm 0.0043}$ | $0.40 \pm 0.80$ |
| | 80% flips | NoTrans | $\mathbf{17.52 \pm 0.60}$ | $\mathbf{0.8252 \pm 0.0059}$ | $\mathbf{0.1663 \pm 0.0047}$ | $\mathbf{0.60 \pm 0.49}$ |
| | | LinearProbe | $1.00 \pm 0.00$ | $0.6740 \pm 0.0019$ | $0.0002 \pm 0.0000$ | $0.00 \pm 0.00$ |
| | | Adapter | $1.00 \pm 0.00$ | $0.5250 \pm 0.0058$ | $0.0002 \pm 0.0000$ | $0.00 \pm 0.00$ |
| | | Distill | $15.11 \pm 0.49$ | $0.8174 \pm 0.0069$ | $0.1227 \pm 0.0039$ | $0.00 \pm 0.00$ |
| | | LoRA | $2.01 \pm 0.18$ | $0.6251 \pm 0.0032$ | $0.0026 \pm 0.0006$ | $0.00 \pm 0.00$ |
| | | DANN-Gate | $1.00 \pm 0.00$ | $0.5754 \pm 0.0113$ | $0.0002 \pm 0.0000$ | $0.00 \pm 0.00$ |
| | | REFINE | $17.37 \pm 1.09$ | $0.8239 \pm 0.0060$ | $0.1641 \pm 0.0109$ | $0.20 \pm 0.40$ |
| | Schematic confusion | NoTrans | $18.13 \pm 0.74$ | $0.8129 \pm 0.0044$ | $0.1747 \pm 0.0073$ | $1.20 \pm 0.75$ |
| | | LinearProbe | $20.81 \pm 0.13$ | $0.8316 \pm 0.0003$ | $0.2006 \pm 0.0038$ | $0.60 \pm 0.80$ |
| | | Adapter | $19.99 \pm 0.24$ | $0.8308 \pm 0.0012$ | $0.1895 \pm 0.0052$ | $0.00 \pm 0.00$ |
| | | Distill | $20.06 \pm 0.89$ | $\mathbf{0.8361 \pm 0.0077}$ | $0.1959 \pm 0.0080$ | $1.00 \pm 0.63$ |
| | | LoRA | $20.05 \pm 0.18$ | $0.8246 \pm 0.0017$ | $0.1953 \pm 0.0035$ | $0.60 \pm 0.80$ |
| | | DANN-Gate | $17.56 \pm 0.33$ | $0.8122 \pm 0.0023$ | $0.1720 \pm 0.0032$ | $0.00 \pm 0.00$ |
| | | REFINE | $\mathbf{21.76 \pm 0.60}$ | $0.8308 \pm 0.0072$ | $\mathbf{0.2139 \pm 0.0067}$ | $\mathbf{2.00 \pm 1.10}$ |
| | Class imbalance | NoTrans | $17.58 \pm 0.24$ | $0.8271 \pm 0.0033$ | $0.1656 \pm 0.0046$ | $\mathbf{1.00 \pm 0.00}$ |
| | | LinearProbe | $22.41 \pm 0.48$ | $0.8687 \pm 0.0011$ | $0.2133 \pm 0.0048$ | $0.00 \pm 0.00$ |
| | | Adapter | $22.66 \pm 0.30$ | $0.8676 \pm 0.0014$ | $0.2102 \pm 0.0025$ | $0.00 \pm 0.00$ |
| | | Distill | $19.59 \pm 0.61$ | $0.8659 \pm 0.0034$ | $0.1752 \pm 0.0072$ | $0.00 \pm 0.00$ |
| | | LoRA | $22.56 \pm 0.39$ | $0.8535 \pm 0.0009$ | $0.2129 \pm 0.0022$ | $0.00 \pm 0.00$ |
| | | DANN-Gate | $20.72 \pm 0.24$ | $0.8432 \pm 0.0021$ | $0.1966 \pm 0.0031$ | $0.00 \pm 0.00$ |
| | | REFINE | $\mathbf{23.31 \pm 0.42}$ | $\mathbf{0.8719 \pm 0.0010}$ | $\mathbf{0.2264 \pm 0.0032}$ | $0.40 \pm 0.49$ |

Table S1: Single-source transfer learning with label noise, semantic perturbation, and class imbalance for CIFAR-100 using CNNs.

| Dataset | Setting | Method | Acc | AUC | F1 | MinCAcc |
|---------|---------|--------|-----|-----|-----|---------|
| CIFAR-10 | 80% flips | NoTrans | $45.17 \pm 1.39$ | $0.8678 \pm 0.0028$ | $0.4391 \pm 0.0183$ | $16.24 \pm 4.52$ |
| | | LinearProbe | $20.65 \pm 0.44$ | $0.6826 \pm 0.0025$ | $0.1410 \pm 0.0083$ | $0.00 \pm 0.00$ |
| | | Adapter | $17.88 \pm 0.73$ | $0.6682 \pm 0.0066$ | $0.1248 \pm 0.0111$ | $0.00 \pm 0.00$ |
| | | Distill | $40.19 \pm 0.57$ | $0.8445 \pm 0.0022$ | $0.3827 \pm 0.0068$ | $8.00 \pm 5.22$ |
| | | LoRA | $21.69 \pm 0.49$ | $0.6831 \pm 0.0010$ | $0.1511 \pm 0.0059$ | $0.00 \pm 0.00$ |
| | | DANN-Gate | $21.37 \pm 0.27$ | $0.6829 \pm 0.0015$ | $0.1468 \pm 0.0075$ | $0.00 \pm 0.00$ |
| | | REFINE | $\mathbf{45.53 \pm 0.95}$ | $\mathbf{0.8694 \pm 0.0047}$ | $\mathbf{0.4463 \pm 0.0105}$ | $\mathbf{18.68 \pm 4.97}$ |
| | Domain mismatch | NoTrans | $44.37 \pm 0.74$ | $0.8628 \pm 0.0035$ | $0.4375 \pm 0.0055$ | $20.80 \pm 4.86$ |
| | | LinearProbe | $46.04 \pm 0.71$ | $0.8643 \pm 0.0015$ | $0.4544 \pm 0.0080$ | $23.46 \pm 4.74$ |
| | | Adapter | $44.87 \pm 0.55$ | $0.8514 \pm 0.0029$ | $0.4445 \pm 0.0059$ | $26.74 \pm 1.89$ |
| | | LoRA | $47.74 \pm 0.38$ | $\mathbf{0.8752 \pm 0.0015}$ | $\mathbf{0.4750 \pm 0.0032}$ | $27.96 \pm 2.61$ |
| | | DANN-Gate | $\mathbf{47.79 \pm 0.40}$ | $0.8750 \pm 0.0019$ | $0.4733 \pm 0.0036$ | $28.12 \pm 4.52$ |
| | | REFINE | $44.85 \pm 0.38$ | $0.8524 \pm 0.0011$ | $0.4474 \pm 0.0035$ | $\mathbf{29.68 \pm 1.78}$ |
| | Schematic confusion | NoTrans | $45.36 \pm 0.59$ | $0.8662 \pm 0.0033$ | $0.4455 \pm 0.0081$ | $18.98 \pm 7.49$ |
| | | LinearProbe | $53.45 \pm 0.44$ | $0.9090 \pm 0.0002$ | $0.5259 \pm 0.0078$ | $26.28 \pm 6.59$ |
| | | Adapter | $52.67 \pm 0.33$ | $0.9089 \pm 0.0008$ | $0.5195 \pm 0.0050$ | $30.84 \pm 4.96$ |
| | | Distill | $46.01 \pm 1.11$ | $0.8736 \pm 0.0028$ | $0.4435 \pm 0.0143$ | $14.00 \pm 6.94$ |
| | | LoRA | $52.35 \pm 0.42$ | $0.9024 \pm 0.0008$ | $0.5176 \pm 0.0053$ | $32.50 \pm 0.97$ |
| | | DANN-Gate | $52.13 \pm 0.35$ | $0.9021 \pm 0.0009$ | $0.5141 \pm 0.0036$ | $33.28 \pm 4.33$ |
| | | REFINE | $\mathbf{54.62 \pm 0.45}$ | $\mathbf{0.9134 \pm 0.0010}$ | $\mathbf{0.5431 \pm 0.0056}$ | $\mathbf{33.90 \pm 3.34}$ |
| | Class imbalanace | NoTrans | $45.36 \pm 1.39$ | $0.8678 \pm 0.0028$ | $0.4391 \pm 0.0183$ | $16.24 \pm 4.52$ |
| | | LinearProbe | $48.44 \pm 0.37$ | $0.8749 \pm 0.0008$ | $0.4805 \pm 0.0052$ | $25.94 \pm 6.98$ |
| | | Adapter | $47.57 \pm 0.27$ | $0.8678 \pm 0.0029$ | $0.4689 \pm 0.0045$ | $25.26 \pm 4.53$ |
| | | Distill | $42.25 \pm 0.63$ | $0.8650 \pm 0.0035$ | $0.3996 \pm 0.0051$ | $3.86 \pm 0.82$ |
| | | LoRA | $\mathbf{48.99 \pm 0.30}$ | $0.8759 \pm 0.0007$ | $\mathbf{0.4866 \pm 0.0036}$ | $30.92 \pm 3.71$ |
| | | DANN-Gate | $48.94 \pm 0.41$ | $\mathbf{0.8766 \pm 0.0009}$ | $0.4860 \pm 0.0051$ | $\mathbf{31.62 \pm 1.52}$ |
| | | REFINE | $47.81 \pm 0.23$ | $0.8691 \pm 0.0007$ | $0.4755 \pm 0.0026$ | $29.44 \pm 3.26$ |

Table S2: Single-source transfer learning with label noise, semantic perturbation, and class imbalance for CIFAR-10 using transformers.

## C.4 TABULAR DATA

We demonstrate that REFINE is equally effective in handling tabular data. We consider three binary-class datasets, Adult [23], Credit [35], Diabetes [44], and one multi-class dataset, Performance [43].

| Dataset | Setting | Method | Acc | AUC | F1 | MinCAcc |
|---|---|---|---|---|---|---|
| CIFAR-100 | 80% flips | NoTrans | $15.32 \pm 0.33$ | $\mathbf{0.8449 \pm 0.0021}$ | $0.1358 \pm 0.0041$ | $0.00 \pm 0.00$ |
| | | LinearProbe | $6.70 \pm 0.27$ | $0.7377 \pm 0.0011$ | $0.0390 \pm 0.0014$ | $0.00 \pm 0.00$ |
| | | Adapter | $6.54 \pm 0.16$ | $0.7405 \pm 0.0011$ | $0.0348 \pm 0.0009$ | $0.00 \pm 0.00$ |
| | | Distill | $11.83 \pm 0.26$ | $0.8130 \pm 0.0027$ | $0.0835 \pm 0.0024$ | $0.00 \pm 0.00$ |
| | | LoRA | $6.97 \pm 0.07$ | $0.7390 \pm 0.0015$ | $0.0428 \pm 0.0014$ | $0.00 \pm 0.00$ |
| | | DANN-Gate | $6.91 \pm 0.23$ | $0.7392 \pm 0.0016$ | $0.0429 \pm 0.0014$ | $0.00 \pm 0.00$ |
| | | REFINE | $\mathbf{15.50 \pm 0.79}$ | $0.8437 \pm 0.0041$ | $\mathbf{0.1378 \pm 0.0067}$ | $0.00 \pm 0.00$ |
| | Domain mismatch | NoTrans | $11.28 \pm 0.52$ | $0.8023 \pm 0.0034$ | $0.0984 \pm 0.0033$ | $0.00 \pm 0.00$ |
| | | LinearProbe | $13.32 \pm 0.52$ | $0.8186 \pm 0.0015$ | $0.1175 \pm 0.0049$ | $0.00 \pm 0.00$ |
| | | Adapter | $12.64 \pm 0.32$ | $0.8267 \pm 0.0006$ | $0.1052 \pm 0.0030$ | $0.00 \pm 0.00$ |
| | | LoRA | $14.22 \pm 0.26$ | $0.8466 \pm 0.0010$ | $0.1289 \pm 0.0028$ | $0.00 \pm 0.00$ |
| | | DANN-Gate | $14.08 \pm 0.37$ | $0.8465 \pm 0.0012$ | $0.1280 \pm 0.0023$ | $0.00 \pm 0.00$ |
| | | REFINE | $\mathbf{14.38 \pm 0.54}$ | $0.8291 \pm 0.0032$ | $\mathbf{0.1329 \pm 0.0039}$ | $0.00 \pm 0.00$ |
| | Schematic confusion | NoTrans | $\mathbf{16.24 \pm 0.58}$ | $\mathbf{0.8471 \pm 0.0036}$ | $\mathbf{0.1485 \pm 0.0075}$ | $0.00 \pm 0.00$ |
| | | LinearProbe | $11.88 \pm 0.28$ | $0.7950 \pm 0.0015$ | $0.1067 \pm 0.0015$ | $0.00 \pm 0.00$ |
| | | Adapter | $11.17 \pm 0.43$ | $0.7936 \pm 0.0027$ | $0.0918 \pm 0.0040$ | $0.00 \pm 0.00$ |
| | | Distill | $15.01 \pm 0.64$ | $0.8266 \pm 0.0028$ | $0.1260 \pm 0.0081$ | $0.00 \pm 0.00$ |
| | | LoRA | $11.36 \pm 0.18$ | $0.7899 \pm 0.0013$ | $0.0991 \pm 0.0015$ | $0.00 \pm 0.00$ |
| | | DANN-Gate | $11.46 \pm 0.21$ | $0.7893 \pm 0.0013$ | $0.0989 \pm 0.0017$ | $0.00 \pm 0.00$ |
| | | REFINE | $14.94 \pm 0.49$ | $0.8282 \pm 0.0026$ | $0.1402 \pm 0.0026$ | $0.00 \pm 0.00$ |
| | Class imbalance | NoTrans | $15.43 \pm 0.32$ | $0.8474 \pm 0.0025$ | $0.1386 \pm 0.0012$ | $0.00 \pm 0.00$ |
| | | LinearProbe | $\mathbf{25.82 \pm 0.28}$ | $0.8877 \pm 0.0010$ | $\mathbf{0.2529 \pm 0.0020}$ | $3.60 \pm 0.80$ |
| | | Adapter | $24.48 \pm 0.32$ | $0.8847 \pm 0.0010$ | $0.2320 \pm 0.0027$ | $0.60 \pm 0.80$ |
| | | Distill | $16.01 \pm 0.13$ | $0.8721 \pm 0.0017$ | $0.1252 \pm 0.0021$ | $0.00 \pm 0.00$ |
| | | LoRA | $23.52 \pm 0.09$ | $0.8669 \pm 0.0015$ | $0.2250 \pm 0.0023$ | $0.00 \pm 0.00$ |
| | | DANN-Gate | $23.48 \pm 0.13$ | $0.8671 \pm 0.0018$ | $0.2264 \pm 0.0019$ | $0.00 \pm 0.00$ |
| | | REFINE | $25.54 \pm 0.43$ | $\mathbf{0.8879 \pm 0.0013}$ | $0.2524 \pm 0.0039$ | $\mathbf{4.80 \pm 0.75}$ |

Table S3: Single-source transfer learning with label noise, semantic perturbation, and class imbalance for CIFAR-100 using transformers.

| Dataset | Metric | Classifier | | | | | |
|---|---|---|---|---|---|---|---|
| | | MLP1 | | | MLP2 | | |
| | | Raw | DirectAug | REFINE | Raw | DirectAug | REFINE |
| Adult | Accuracy | $0.807 \pm 0.008$ | $0.831 \pm 0.006$ | $0.821 \pm 0.004$ | $0.800 \pm 0.011$ | $0.833 \pm 0.005$ | $0.814 \pm 0.010$ |
| | AUC | $0.832 \pm 0.008$ | $0.878 \pm 0.006$ | $0.852 \pm 0.008$ | $0.833 \pm 0.010$ | $0.883 \pm 0.005$ | $0.854 \pm 0.008$ |
| | F1 | $0.547 \pm 0.037$ | $0.619 \pm 0.015$ | $0.595 \pm 0.030$ | $0.570 \pm 0.028$ | $0.627 \pm 0.021$ | $0.612 \pm 0.021$ |
| Credit | Accuracy | $0.723 \pm 0.028$ | $0.735 \pm 0.017$ | $0.740 \pm 0.022$ | $0.717 \pm 0.027$ | $0.732 \pm 0.015$ | $0.726 \pm 0.020$ |
| | AUC | $0.730 \pm 0.024$ | $0.738 \pm 0.013$ | $0.745 \pm 0.018$ | $0.725 \pm 0.025$ | $0.754 \pm 0.020$ | $0.736 \pm 0.023$ |
| | F1 | $0.490 \pm 0.043$ | $0.524 \pm 0.030$ | $0.520 \pm 0.038$ | $0.515 \pm 0.041$ | $0.541 \pm 0.030$ | $0.535 \pm 0.037$ |
| Diabetes | Accuracy | $0.565 \pm 0.015$ | $0.573 \pm 0.008$ | $0.571 \pm 0.008$ | $0.561 \pm 0.015$ | $0.596 \pm 0.007$ | $0.572 \pm 0.010$ |
| | AUC | $0.582 \pm 0.019$ | $0.597 \pm 0.008$ | $0.591 \pm 0.012$ | $0.576 \pm 0.018$ | $0.626 \pm 0.008$ | $0.593 \pm 0.013$ |
| | F1 | $0.505 \pm 0.028$ | $0.533 \pm 0.014$ | $0.523 \pm 0.022$ | $0.501 \pm 0.029$ | $0.534 \pm 0.017$ | $0.522 \pm 0.027$ |
| Performance | Accuracy | $0.684 \pm 0.019$ | $0.724 \pm 0.011$ | $0.711 \pm 0.014$ | $0.683 \pm 0.018$ | $0.668 \pm 0.084$ | $0.702 \pm 0.022$ |
| | AUC | $0.857 \pm 0.011$ | $0.878 \pm 0.009$ | $0.869 \pm 0.009$ | $0.858 \pm 0.011$ | $0.830 \pm 0.070$ | $0.865 \pm 0.011$ |
| | F1 | $0.478 \pm 0.025$ | $0.557 \pm 0.024$ | $0.521 \pm 0.029$ | $0.478 \pm 0.027$ | $0.494 \pm 0.090$ | $0.507 \pm 0.035$ |

Table S4: Single-source transfer learning with original tabular data.

Each raw training data contains $K \times 100$ samples, where $K$ is the number of classes. To assess model complexity, we design two multilayer perceptron (MLP) architectures: MLP1 with a lower complexity, and MLP2 with a more complex structure. We also compare to DirectAug, which refers to directly combining additional data with the raw data to train the classifier.

Table S4 reports the results using the original data, and Table S5 reports the results using the noisy data with 80% flips of class labels. In both settings, REFINE consistently improves accuracy, AUC, and F1 over using the raw data alone. Although DirectAug can sometimes perform better through full data merging, REFINE surpasses it on several datasets, including Credit and Performance, confirming its ability to exploit useful auxiliary information without over-relying on data merging. In the presence of heavy label noise, DirectAug suffers severe degradation, whereas REFINE maintains or slightly improves performance. Overall, these results show that REFINE is effective on tabular data, and offers a safe and reliable mechanism for leveraging additional data compared to direct augmentation.

| Dataset | Metric | Classifier | | | | | |
|---|---|---|---|---|---|---|---|
| | | MLP1 | | | MLP2 | | |
| | | Raw | DirectAug | REFINE | Raw | DirectAug | REFINE |
| Adult | Accuracy | $0.808 \pm 0.007$ | $0.615 \pm 0.046$ | $0.805 \pm 0.008$ | $0.800 \pm 0.010$ | $0.641 \pm 0.052$ | $0.791 \pm 0.016$ |
| | AUC | $0.834 \pm 0.009$ | $0.612 \pm 0.051$ | $0.832 \pm 0.010$ | $0.834 \pm 0.013$ | $0.639 \pm 0.052$ | $0.828 \pm 0.014$ |
| | F1 | $0.549 \pm 0.046$ | $0.383 \pm 0.039$ | $0.555 \pm 0.029$ | $0.564 \pm 0.032$ | $0.395 \pm 0.047$ | $0.562 \pm 0.027$ |
| Credit | Accuracy | $0.723 \pm 0.027$ | $0.581 \pm 0.035$ | $0.705 \pm 0.028$ | $0.716 \pm 0.027$ | $0.599 \pm 0.045$ | $0.705 \pm 0.028$ |
| | AUC | $0.728 \pm 0.027$ | $0.578 \pm 0.045$ | $0.705 \pm 0.028$ | $0.720 \pm 0.026$ | $0.599 \pm 0.045$ | $0.687 \pm 0.028$ |
| | F1 | $0.483 \pm 0.049$ | $0.417 \pm 0.048$ | $0.481 \pm 0.035$ | $0.512 \pm 0.041$ | $0.433 \pm 0.041$ | $0.493 \pm 0.034$ |
| Diabetes | Accuracy | $0.587 \pm 0.007$ | $0.516 \pm 0.007$ | $0.575 \pm 0.006$ | $0.614 \pm 0.006$ | $0.551 \pm 0.016$ | $0.609 \pm 0.004$ |
| | AUC | $0.580 \pm 0.020$ | $0.516 \pm 0.014$ | $0.554 \pm 0.016$ | $0.577 \pm 0.017$ | $0.585 \pm 0.019$ | $0.567 \pm 0.020$ |
| | F1 | $0.503 \pm 0.032$ | $0.489 \pm 0.025$ | $0.498 \pm 0.022$ | $0.503 \pm 0.026$ | $0.483 \pm 0.037$ | $0.514 \pm 0.025$ |
| Performance | Accuracy | $0.682 \pm 0.020$ | $0.637 \pm 0.088$ | $0.696 \pm 0.023$ | $0.684 \pm 0.018$ | $0.650 \pm 0.079$ | $0.696 \pm 0.023$ |
| | AUC | $0.857 \pm 0.011$ | $0.805 \pm 0.074$ | $0.862 \pm 0.011$ | $0.859 \pm 0.010$ | $0.814 \pm 0.068$ | $0.863 \pm 0.012$ |
| | F1 | $0.476 \pm 0.028$ | $0.464 \pm 0.096$ | $0.499 \pm 0.036$ | $0.480 \pm 0.029$ | $0.472 \pm 0.088$ | $0.500 \pm 0.035$ |

Table S5: Single-source transfer learning with noisy tabular data.

## C.5 Ablation Studies

We conduct an ablation study to investigate the effect of complexity of the encoder $h$ in REFINE, by varying the width and depth of the neural network models used. Figure S3 reports the performance of REFINE under five different models with increasing complexity for $h$. The left panel reports the total number of trainable parameters, the middle panel reports the classification accuracy using the original data, and the right panel using the noisy data. On the original data, REFINE consistently outperforms NoTrans across all levels of complexity by a considerable margin, demonstrating its ability to leverage useful pretrained features. On the noisy data, REFINE performs on par with NoTrans regardless of the complexity of $h$, confirming its robustness to negative transfer. Together, these results show that REFINE offers robust and reliable safeguarding against negative transfer.

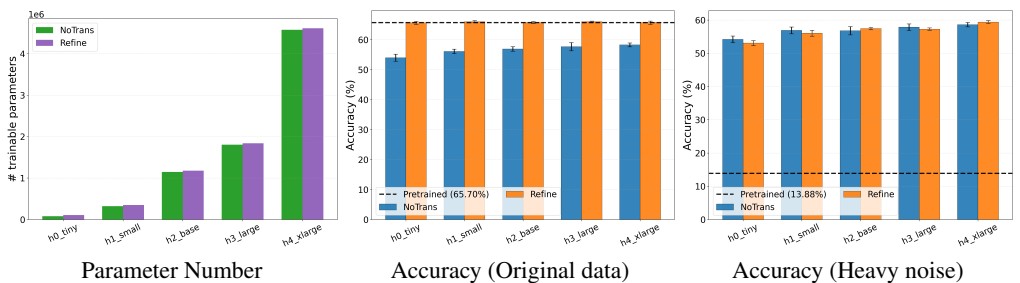

Parameter Number     Accuracy (Original data)     Accuracy (Heavy noise)

Figure S3: Ablation study for the encoder $h$ with varying complexity.

A related ablation on adapter size further confirms that negative transfer is not due to insufficient parameter count. Here, 1x corresponds to the same adapter size used in the main experiment in Table 1. As shown in Fig. S4, enlarging the adapter from 1x to 500x yields only minor fluctuations around 65–66 percent accuracy, 0.72 AUC, and 0.65 F1, and never approaches the NoTrans baseline at 68.5 percent accuracy or 0.76 AUC. In contrast, REFINE reaches 70.3 percent accuracy and 0.79 AUC, clearly surpassing both NoTrans and all adapter scales. These results show that increasing capacity alone cannot overcome the source–target mismatch responsible for negative transfer, while REFINE remains the only mechanism that reliably corrects it.

## D More Details on Experiment Setup and Implementations

We provide additional details on experiment setup and implementations for better reproducibility. All experiments are conducted on an NVIDIA A10G (Ampere) GPU with 23 GB of GDDR6 memory, driver version 535.183.01, and CUDA 12.2. For semantic confusion in CIFAR-10 and CIFAR-100, we construct 4 and 47 pairs of related classes, respectively, and flip $50\%$ of each pair's samples to its counterpart, while also injecting white noise into image attributes with $\sigma = 0.2$. For class

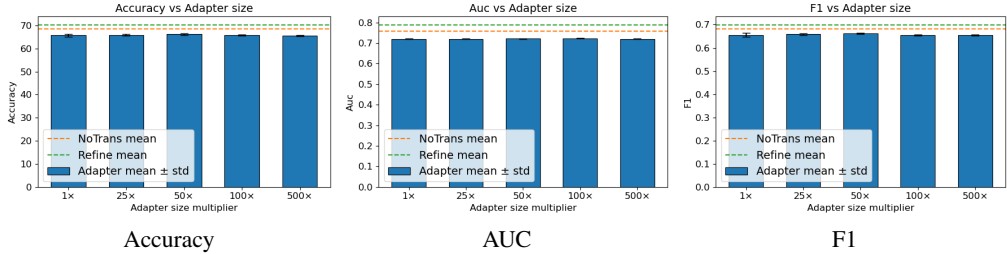

Figure S4: Performance of Adapter under varying parameter count multipliers.

imbalance, we create each imbalanced pretrained subset by first sampling 10,000 images from the full training split with a fixed seed (42). In CIFAR-10, classes 0-9 are sampled with proportions $[0.35, 0.30, 0.10, 0.07, 0.06, 0.045, 0.03, 0.02, 0.015, 0.01]$, yielding 3,500 to 100 images per class. In CIFAR-100, the first 10 classes are designated as majority, with 400 images each, and the remaining 90 as minority, with 100 images each, truncated to a total of 10,000 samples. Table S6 summarizes the experiment settings.

| Dataset | Pretrained Model | Base Model | Pretrain Size | Fine-tune Size | Adapter Para (%) | REFINE Para (%) |
|---|---|---|---|---|---|---|
| CIFAR-CNN-related | CNN | CNN | 10000 | 4000 | 5.46 | 4.88 |
| CIFAR-TF-related | Transformer | Transformer | 10000 | 4000 | 6.49 | 4.63 |
| CIFAR-10→STL | CNN | CNN | 10000 | 4000 | 5.46 | 4.88 |
| Clipart→Sketch | ResNet18 | ResNet10 | 3000 | 1000 | 1.36 | 44.2 |
| USPS→MNIST | CNN | CNN | 5000 | 100 | 5.46 | 4.88 |
| Books→Kitchen | Transformer | Transformer | 2000 | 400 | 2.25 | 96.58 |
| DVD→Electronics | Transformer | Transformer | 2000 | 400 | 2.25 | 96.58 |

Table S6: Experiment settings for all data examples.

We clarify the exact model architectures used. For CNN experiments, the finetuned model is a standard three-block convolutional network with channels $\{32, 64, 64\}$, where each block consists of a $3 \times 3$ convolution (padding 1), ReLU activation, and $2 \times 2$ max pooling, followed by a 512-dimensional fully connected layer and a linear classifier. The pretrained CNN is a larger backbone with convolutional stages $\{80, 160, 320, 640, 640, 768\}$, followed by a 2560-dimensional projection layer and a linear classifier. For transformer experiments, the finetuned model is a lightweight vision transformer with patch size 4, embedding dimension 128, two encoder layers, a 512-dimensional MLP head, and a linear classifier. The pretrained transformer uses patch size 2, embedding dimension 512, six encoder layers, a 2560-dimensional projection head, and a linear classifier. For the DomainNet experiments, following standard practice, we use ResNet-10 as the finetuned model and ResNet-18 (from `torchvision`) as the pretrained model.

## E    FURTHER DISCUSSION ABOUT RELATED WORK

**Transfer learning.**    The affine model transformation (AMT) approach [32] is fundamentally different from our setting. AMT only applies an output-level update of the form $f_T(x) = a \cdot f_S(x) + b$, which corresponds to a global scale and bias correction on the pretrained model. Such a transformation cannot address representation-level mismatch, nonlinear domain shift, or structured encoder errors, nor can it introduce new features or modalities. In contrast, REFINE modifies adaptation at the representation level by keeping the pretrained encoder fixed and introducing an additive residual encoder that corrects the internal representation. This allows the predictor to change its entire functional form rather than merely rescale outputs. The residual structure also provides a natural safety property: when the pretrained model is helpful, the residual remains small; when it is harmful, the residual can override it, yielding performance no worse than training from scratch. AMT does not provide this fallback guarantee and cannot accommodate new modalities, whereas REFINE can incorporate additional sources of information at adaptation time (e.g., spatial encoders added atop scGPT in our spatial-omics experiments).

While the deep transfer learning (DTL) framework [21] also introduces an auxiliary component beyond the base representation, its goals and assumptions differ substantially from ours. DTL retrains the representation using all upstream domains jointly with Wasserstein and distance-covariance penalties, requiring full access to multi-domain source data and a fixed set of domains and modalities during pretraining. Only after this upstream retraining is completed is the target-domain predictor then fitted under an independence constraint. In contrast, REFINE assumes a fixed pretrained model from the outset and introduces a residual encoder *only at adaptation time* to correct the frozen representation on the target distribution. This design enables our no-negative-transfer guarantee and fallback to the target-only estimator—properties not provided by DTL. Moreover, because DTL assumes that no new modalities appear after upstream training, it cannot handle scenarios such as spatial-omics where new sources of information become available exclusively at adaptation time, precisely the regime targeted by REFINE.

**Baseline selection for negative-transfer evaluation.** Our selection of baselines follows recent recommendations from studies on negative transfer (NT) and parameter-efficient fine-tuning (PEFT). Importantly, our goal is not to benchmark raw target accuracy against the newest domain-alignment algorithms, but to evaluate robustness to negative transfer, for which the modern literature identifies only a small set of meaningful baselines. Recent PEFT analyses [31] show that most contemporary PEFT variants behave similarly under distribution shift and provide little or no protection against negative transfer; thus, LoRA serves as a representative and widely used PEFT baseline for NT evaluation. Likewise, the NT survey literature emphasizes that very few modern transfer-learning methods are explicitly designed with safety objectives in mind; accordingly, adversarial domain adaptation approaches such as DANN remain the standard benchmarks used in NT studies [49]. Many newer transfer-learning methods primarily target domain alignment or feature matching but lack any mechanism for safety or fallback, so including additional variants would not meaningfully strengthen the evaluation. Consistent with the NT literature [49], we therefore adopt a baseline set that directly probes safety: NoTrans, feature-based adaptation (LinearProbe and Adapter), adversarial DA (DANN), and one representative PEFT method (LoRA). These baselines provide the appropriate lens for assessing whether REFINE achieves its intended property of avoiding negative transfer rather than merely improving average accuracy.

**Source-free multi-source transfer.** Our multi-source experiment operates under a source-free, adaptation-time setting in which the pretrained model is fixed and no upstream source data are accessible during adaptation. Under this constraint, classical multi-source transfer algorithms that rely on joint training over all source domains, full access to source datasets, re-optimization of a shared encoder, or traditional boosting [48] cannot be applied, including multi-source domain alignment methods, mixture-of-experts training frameworks, and multi-source adversarial domain adaptation approaches. These methods fundamentally assume retraining with all sources present and therefore fall outside the feasible operation regime of our setting. At adaptation time, we only receive a small number of target-like auxiliary sources, often with mismatched structure, and have no ability to revisit any upstream data. Consequently, the only baselines that are valid in this source-free scenario are NoTrans and simple data concatenation. These baselines reflect the operations that a practitioner can realistically perform when upstream data cannot be accessed and isolate the negative-transfer phenomenon that we aim to study, namely how to safely incorporate multiple heterogeneous sources without degrading downstream performance.

