# OpenReview forum: "Residual Feature Integration is Sufficient to Prevent Negative Transfer"
_ICLR.cc/2026/Conference — ICLR 2026 Poster_

### Official Review · Reviewer_cD6H · 2025-10-21

**Soundness:** 2
**Presentation:** 2
**Contribution:** 2
**Rating:** 2
**Confidence:** 4

**Summary:**

This paper studies transfer learning and proposes a residual feature integration method to mitigate negative transfer. The authors also introduce an upper bound for the generalization error under a nonparametric regression setup. However, the theoretical contribution is somewhat limited, and the empirical comparisons lack depth in several aspects.

**Strengths:**

1.The paper identifies a simple, intuitive, and model-agnostic method (ReFine) to address the persistent problem of negative transfer. The strategy of augmenting a frozen pre-trained feature extractor with a trainable residual encoder is straightforward to implement and integrates easily into existing transfer learning pipelines.
2.The work provides a rigorous theoretical analysis demonstrating that ReFine provably prevents negative transfer.
3.The method is evaluated across a wide range of settings, including image, text, and tabular data. Experiments cover natural distribution shifts and challenging stress tests (e.g., high label noise, semantic perturbation, class imbalance), consistently showing that ReFine mitigates performance degradation where other methods fail.

**Weaknesses:**

1.While the theoretical analysis is rigorous, the core architecture—a linear combination of a pre-trained feature map and a ReLU network—is structurally similar to models previously analyzed in the deep learning theory literature. The resulting generalization bounds, though well-derived, may not represent a significant conceptual advance over existing approximation theories for neural networks.
2.The selection of baseline methods lacks contemporary relevance. Compared approaches like DANN and LoRA are established but not state-of-the-art. The empirical evaluation would be more compelling if it included more recent and influential transfer learning strategies, providing a clearer view of the method's current competitive standing.
3.The paper does not provide a detailed comparison of the number of trainable parameters across all methods. This omission makes it difficult to assess the true efficiency of ReFine and to perform a fair comparison with parameter-efficient fine-tuning baselines like Adapter and LoRA.
4.The multi-source transfer experiments are evaluated against only two simple baselines (NoTrans and a Naive concatenation). This setup is not sufficient to demonstrate a clear advantage, as it lacks comparison with dedicated and potentially stronger multi-source transfer learning algorithms.

**Questions:**

1. The proposed model, \hat{g}(x)=v⊤frep(x)+uh(x) can be viewed as a ReLU network directly according to the descriptions on P5 . The approximation analysis for such architectures has been extensively studied in prior works, and the theoretical guarantees provided here do not appear to deviate significantly from existing results.
2.The baseline methods selected for comparison—such as DANN and LoRA—are relatively outdated. Several influential and more recent transfer learning strategies are not included, which limits the relevance and comprehensiveness of the experimental evaluation.
3. A detailed comparison of the number of parameters across different methods is missing. This makes it difficult to assess the efficiency of the proposed approach relative to existing alternatives.
4. In the multi-source transfer experiments, only two simple baselines (NoTrans and Naive) are used. This is insufficient to demonstrate the advantages of the proposed method in multi-source settings, where more competitive or state-of-the-art approaches should be considered.

---

> ### Author Response · Authors · 2025-12-02
>
> - [Difference with the existing literature.]
> We agree that the architecture $\hat g(x) = v^T f_{rep}(x) + u h(x)$ can be viewed as a ReLU network. Our contribution is not to obtain sharper approximation results for general ReLU networks, but to establish safe transfer guarantees and representation-adaptive excess-risk bounds not captured by standard approximation analyses.
>
> Classical ReLU theory focuses on a single inductive bias and does not compare estimators across two competing biases: linear prediction on a pretrained representation $f_{rep}$ versus nonparametric learning from raw target inputs. Our analysis does so. In particular, Theorem~4.1 yields the representation-adaptive excess-risk bound $$E[R_{P^t}(\hat g) - R_{P^t}(f^*)] = \tilde O\left(\rho^{2 d/(2 \beta + d)} n^{-2 \beta/(2 \beta + d)} + \frac{p}{n}\right).$$
>
> Corollary~4.4 establishes a no-negative-transfer guarantee: $$\sup_{f^{\star} \in F^{\beta}(f_{rep}, \gamma)} E[R_{P^{t}}(\hat g) - R_{P^{t}}(f^{\star})] = \tilde O(\min(\sup_{f^{\star} \in F^{\beta}(f_{rep}, \gamma)} E[R_{P^{t}}(\hat g_{sc}) - R_{P^{t}}(f^{\star})], \sup_{f^{\star} \in F^{\beta}(f_{rep}, \gamma)} E[R_{P^{t}}(\hat w_{ft}^{T} f_{rep}) - R_{P^{t}}(f^{\star})])).$$
>
> The function class is $$F^{\beta}(f_{rep}, \gamma) := \{ f^* : [0,1]^d \to \mathbb{R} \mid \min_{v : |v| \le 1} | v^T f_{rep} - f^* |_{C^{\beta}} \le \gamma \}.$$
>
>
> - [Baseline methods] Thank you for raising this concern. Our baseline selection follows recent negative transfer (NT) and PEFT literature. Our goal is not to benchmark against the newest domain-alignment methods, but to evaluate safety against NT, for which a small, established set of baselines is standard. Recent work [1] shows that most PEFT variants behave similarly under distribution shift and do not prevent NT, making LoRA a representative PEFT baseline. Likewise, NT surveys highlight that few transfer methods explicitly address safety, so DANN remains the standard adversarial adaptation baseline [2]. Many newer approaches focus on alignment without safety mechanisms, so including them would not strengthen the evaluation. Following prior NT benchmarks [2], we therefore use NoTrans, LinearProbe, Adapter, DANN, and LoRA to directly assess whether REFINE achieves its intended goal of avoiding negative transfer rather than merely improving accuracy.
>
> [1] Lessons and Insights from a Unifying Study of Parameter-Efficient Fine-Tuning (PEFT) in Visual Recognition, CVPR 25
>
> [2] A Survey on Negative Transfer, IEEE/CAA Journal of Automatica Sinica 23
>
> - [Comment 3: trainable parameters] Thank you for raising this point. We agree that parameter efficiency is important and provide a detailed comparison in the revision. Our results show that negative transfer cannot be eliminated merely by increasing trainable parameters, as summarized in Table S6 and Figs. S3–S4. Table S6 reports the trainable-parameter ratios for Adapter and REFINE across all benchmarks, showing that REFINE remains effective over a wide range of model sizes, from very lightweight residual encoders to larger ones. Figure S3 further confirms that varying the width and depth of the REFINE encoder (and thus its parameter budget) does not compromise robustness to negative transfer. In contrast, Fig. S4 shows that scaling Adapter capacity from 1× up to 500× fails to resolve negative transfer, with performance plateauing around 65–66% accuracy and ~0.72 AUC, never reaching even the NoTrans baseline (68.5% accuracy, 0.76 AUC). Thus, although Adapter and LoRA have well-defined parameter budgets, increasing their capacity does not mitigate negative transfer, whereas REFINE succeeds even under comparable or smaller budgets. These results demonstrate that negative transfer is not a capacity bottleneck but requires REFINE’s residual correction mechanism to address representation mismatch rather than relying on raw parameter scaling.
>
> - [Comment 4: Multi-source Transfer] Thank you for the comment. Our multi-source experiment is conducted in a source-free, adapt-time setting where the pretrained model is fixed and no upstream source data are accessible during adaptation. Under this constraint, classical multi-source transfer methods that require joint training over all sources, full access to source datasets, or re-optimizing a shared encoder (e.g., multi-source domain alignment, mixture-of-experts, adversarial DA) and traditional boosting [3] are inapplicable. At adaptation time, we only receive a few small target-like auxiliary sources and cannot revisit any upstream data. Therefore, the only feasible baselines are NoTrans and simple concatenation, which reflect what a practitioner can actually deploy in this setting and isolate the negative-transfer effect of combining heterogeneous sources. We clarify this setting and rationale in Section C.1 of the revision.
>
> [3] Boosting for transfer learning with multiple sources, CVPR 10

---

### Official Review · Reviewer_fQBT · 2025-10-23

**Soundness:** 3
**Presentation:** 3
**Contribution:** 1
**Rating:** 4
**Confidence:** 4

**Summary:**

This paper introduces a transfer learning method called REFINE for mitigating the phenomenon of negative transfer. Specifically, the authors employ an additional term in addition to composing a linear probe based on the learned representation from the source domain. They provide several experiments to demonstrate the effectiveness of the proposed method. Moreover, a upper bound of excess risk regarding fine-tuning is also offered by this study.

**Strengths:**

1. This study is well-written and easy to follow. The motivation behind their method is reasonable.
2. They conducted comprehensive transfer learning experiments to demonstrate the effectiveness of their method in transfer learning scenarios, which are sound.
3. Meanwhile, the theoretical results obtained by this work are rigorous.

**Weaknesses:**

1. The core idea of this paper is similar to that of the following papers. Both add extra terms to model the potential differences between the source and target tasks that are not successfully captured by the learned representation, in addition to the linear probe on the learned encoder. I sincerely suggest that the authors discuss the differences between their work and these studies in the related works section.

[1] Transfer learning with affine model transformation. Advances in Neural Information Processing Systems, 36, 17296-17329.

[2] Deep transfer learning: Model framework and error analysis. arXiv preprint arXiv:2410.09383.

2. Some of the presented accuracy results seem a little bit weird. For example, the results transferring from CIFAR100 to CIFAR10 are lower than expected. Based on my experience, it should be around 80%, rather than the figure given in the paper (around 40%). The reason for this phenomenon seems to be that they only adopt 10,000 as the pretraining size. I don't understand the rationale behind such a design. Can the authors elaborate on that? This raises a concern about whether the conclusion still holds under sufficient pretraining.

3. The experiments conducted in this study are all focused on small-scale datasets. But I am totally okay with that, given that they have a solid theoretical guarantee.

**Questions:**

The authors claimed that they desire to learn $h$ and $w$ such that
$$
\mathcal{R}\_{\mathbb{P}^t}(w \circ (f\_{rep}, h)) \leq \min\\{\mathcal{R}\_{\mathbb{P}^t}(w_{ft} \circ f_{rep}), \mathcal{R}\_{\mathbb{P}^t}(g\_{sc})\\}
$$
I wonder did authors demonstrate this holds for ERM $(\hat{w}, \hat{h})$ at somewhere of this paper? Or this is just a desire? If this can be justified, what kind of requirements are necessary for the risk $\mathcal{L}$?

**Summary of review:** I think this is a solid paper overall, considering its comprehensive and sound experiments as well as the theoretical guarantees provided. However, the originality of the idea is not clear, along with some experimental concerns mentioned above. Therefore, I would like to set my initial rating at 4. If author successfully resolve my concerns, I will be pleasure to raise my rating.

---

> ### Author Response · Authors · 2025-12-02
>
> We appreciate the reviewer’s constructive feedback and respond to the comments point by point below.
>
> - [Comparison with related works 1] The affine model-transformation (AMT) method is fundamentally different from our setting. It applies only an output-level update $f_T(x)=a f_S(x)+b$, i.e., a global rescaling and bias shift, which cannot address representation mismatch, nonlinear domain shift, encoder errors, or introduce new features or modalities. REFINE instead keeps the pretrained encoder fixed and adds a residual encoder that corrects internal representations, allowing changes to the full predictive function. This yields a safety property: the residual stays small when the source model is helpful and overrides it when harmful, ensuring no-worse-than-scratch performance. AMT lacks this fallback and cannot incorporate new modalities, while REFINE can (e.g., spatial encoders atop scGPT).
>
> - [Comparison with related works 2] DTL also adds a component to representations, but differs in setting and goals. DTL retrains representations across source domains using Wasserstein and distance-covariance penalties, requiring full access to multi-domain data, and fits the target model only after upstream training under independence constraints. REFINE instead assumes a fixed pretrained encoder and adds a residual encoder at adaptation time to correct the target representation. This enables REFINE’s no-negative-transfer guarantee and fallback to target-only training. DTL lacks this residual correction and assumes fixed domains/modalities during pretraining, preventing adaptation-time incorporation of new modalities, unlike REFINE (e.g., spatial-omics).
> In light of these comments, we will add this discussion to Section E of the revision.
>
> - [Experimental results and CIFAR100→CIFAR10 accuracy] Thank you for raising this question. The lower CIFAR100→CIFAR10 accuracies stem from our experimental design rather than a limitation of REFINE. We intentionally pretrain a lightweight CNN on a 10,000-image subset so that repeated pretraining over five seeds is feasible and synthetic stress conditions such as label flips, semantic corruption, and class imbalance can be precisely controlled (Table 2). While state-of-the-art models pretrained on full CIFAR100 can reach around 80% accuracy, our goal is not to optimize absolute performance but to isolate negative-transfer mechanisms under controlled perturbations. Within this regime, REFINE consistently removes negative transfer. Our conclusions are not limited to small CNNs: in the large-scale scGPT spatial-omics experiment, the frozen backbone is pretrained on millions of RNA profiles, yet standard baselines still suffer negative transfer (LinearProbe and Adapter achieve only 0.24 to 0.29 F1 at 1,000 labels versus 0.47 for a target-only GNN). REFINE integrates the unseen spatial modality at adaptation and improves F1 to about 0.52 with 1,000 labels and above 0.70 with 3,000 labels, with higher AUC across all sample sizes.
>
> - [Small-scale datasets] Thank you for the positive assessment. We agree that many experiments use small- to medium-scale datasets, which is a deliberate choice to study negative transfer under tightly controlled conditions with systematic manipulation of label noise, semantic perturbation, and class imbalance. Our theory is scale-independent and does not depend on dataset size or architecture: REFINE is guaranteed to be no worse than target-only training and to benefit from informative representations at any scale. Empirically, we cover diverse modalities (image, text, tabular), CNNs and transformers, and include a large-model scGPT example with adapt-time multimodality, demonstrating that both the mechanism and guarantees extend beyond small-scale settings.
>
> - [Clarification on the question] We thank the reviewer for this comment. We clarify that Theorem 4.1 together with Corollaries 4.2 and 4.3 already provides the comparison between $\hat g$ and the two baselines (“target-only training’’ and “fine-tuning $w \circ f_{rep}$”). Theorem 4.1 establishes the excess-risk rate of $\hat g$. Corollary 4.2 shows that $n^{-2\beta/(2\beta+d)}$ is the minimax-optimal rate using target data only, while by definition $\rho^{*2}$ is the best achievable rate using $w \circ f_{rep}$. Corollary 4.3 further shows that, with properly chosen tuning parameters and fixed $p$, our method achieves the rate $\tilde O\left(\rho^{2d/(2\beta+d)} n^{-2\beta/(2\beta+d)}\right)$, which is no slower than either $n^{-2\beta/(2\beta+d)}$ or $\rho^{*2}$ alone, i.e., the best rates attained by the respective baselines.
>
>
> To make this clearer, in the revision, we have substantially clarified the theoretical statement and provided results that directly establish the claimed no-negative-transfer guarantee.

---

### Official Review · Reviewer_PRLw · 2025-10-26

**Soundness:** 2
**Presentation:** 2
**Contribution:** 2
**Rating:** 2
**Confidence:** 4

**Summary:**

The paper proposes a transfer learning algorithm designed to prevent negative transfer. If the source model provides useful information, the algorithm leverages it; otherwise, its performance does not fall below that of training solely on the target data. It assumes the existence of a pretrained source model $f_{\text{rep}}$. A residual trainable network $h$ is then added, and a linear trainable model $w$ is applied on top of the concatenated representation  $(f_{\text{rep}}(x), h(x))$.

**Strengths:**

The paper derives an error bound for the proposed algorithm, comparing its error to that of the ground truth regression function. Furthermore, the authors conduct numerical experiments demonstrating that their approach is not affected by negative transfer.

**Weaknesses:**

1. The paper claims that its approach avoids negative transfer and that Equation (1) holds for their method, stating that the error of the proposed approach is less than the minimum of the errors obtained by using only target data and by training on top of the source model $f_{\text{rep}}$. However, the theorem they provide does not justify this claim. In fact, Theorem 4.1 only establishes a rate guarantee relative to $f^*$, not an inequality comparing $\hat{g}$—the algorithm’s output—with $\min \\{\text{only target}, w \circ f_{\text{rep}}\\}$. In particular, Theorem 1 does not imply this claim. Therefore, the main claim of the paper remains unproven.

2. A major difficulty with the algorithm lies in tuning the hyperparameter $\rho$, yet the paper provides no method for adaptively selecting this parameter.

**Questions:**

What is the practical justification for the assumption that $||v^*||\leq 1$ ?

---

> ### Author Response · Authors · 2025-12-02
>
> We thank the reviewer for the thoughtful feedback and address each comment in detail below.
>
> - [Clarification over theory] We thank the reviewer for this comment. We clarify that Theorem 4.1 together with Corollaries 4.2 and 4.3 already provides comparisons between $\hat g$ and the two baselines (target-only training and fine-tuning $w \circ f_{rep}$). Theorem 4.1 establishes the excess-risk rate of $\hat g$. Corollary 4.2 shows $n^{-2\beta/(2\beta+d)}$ is the minimax-optimal rate using target data only, while $\rho^{*2}$ is the best achievable rate using $w \circ f_{rep}$. Corollary 4.3 further shows that with properly chosen tuning parameters and fixed $p$ our method achieves
>
> $$
> \tilde O\big(\rho^{2d/(2\beta+d)} n^{-2\beta/(2\beta+d)}\big),
> $$
>
> which is no slower than either baseline alone.
>
> In the revision we state our goal as learning the encoder $\hat h$ and adapter $\hat w$ such that the excess risk of $\hat w \circ (f_{rep},\hat h)$ is bounded by the minimum excess risk of the two baselines $\hat w_{ft}\circ f_{rep}$ and $\hat g_{sc}$.
>
> We define the approximating function class
>
> $$ \mathcal{F}^\beta(f_{rep},\gamma) = \{ f^* : [0,1]^d \to \mathbb{R} \mid \min_{\|v\|\le 1} \| v^\top f_{rep} - f^* \|_{\mathcal{C}^\beta} \le \gamma \}. $$
>
>
> In Corollary 4.4 we introduce
>
> $$
> A =
> \sup_{f^* \in \mathcal{F}^\beta(f_{rep},\gamma)}
> E[ R_t(\hat g) - R_t(f^*) ],
> $$
>
> $$
> A_{sc} =
> \sup_{f^* \in \mathcal{F}^\beta(f_{rep},\gamma)}
> E[ R_t(\hat g_{sc}) - R_t(f^*) ],
> $$
>
> $$
> A_{ft} =
> \sup_{f^* \in \mathcal{F}^\beta(f_{rep},\gamma)}
> E[ R_t(\hat w_{ft}^\top f_{rep}) - R_t(f^*) ].
> $$
>
> These yield the worst-case no-negative-transfer guarantee
>
> $$
> A = \tilde O\big( \min(A_{sc},A_{ft}) \big).
> $$
>
> Finally, Proposition A.1 establishes the asymptotic guarantee
>
> $$
> E[ R_t(\hat w\circ(f_{rep},\hat h)) - R_t(f^*) ]
> \le (1+o(1))\min(B_{ft},B_{sc}) + o(1),
> $$
>
> where
>
> $$
> B_{ft} =
> E[ R_t(\hat w_{ft}^\top f_{rep}) - R_t(f^*) ],
> $$
>
> $$
> B_{sc} =
> E[ R_t(\hat g_{sc}) - R_t(f^*) ].
> $$
>
> This holds under the growth conditions
>
> $$
> W\log(nBL(W+1)L) = o(n),
> \qquad
> p\log n = o(n).
> $$
>
> All proofs are provided in Appendix A.2 (“Worst-Case No Negative Transfer Guarantee”) and Appendix A.3 (“Asymptotic No Negative Transfer Guarantee”), fully addressing the reviewer’s concern.
>
> - [Tuning $\rho$] In practice, tuning $\rho$ is straightforward because REFINE is fully model-agnostic. The user can make hhh larger or smaller, or replace it entirely with any architecture that trains well from scratch on the target domain, and then select ρ using standard validation procedures such as early stopping or a small grid search. A practical default is to choose h with capacity comparable to a scratch model for the target task, which typically yields stable performance. This flexibility also enables REFINE to seamlessly incorporate new modalities at adaptation time; for example, in the spatial-omics experiment, we attach a modality-specific encoder to scGPT without requiring any change to the underlying source model.
>
> - [Practical justification for  $\| v^* \| \leq 1$] Thank you for raising this point. The assumption $\|v^*\| \leq 1$ was made purely for notational simplicity and is not essential to any part of the analysis. In the revised version, we have removed this assumption and restated all relevant results without it.

---

### Official Review · Reviewer_VFA3 · 2025-10-31

**Soundness:** 2
**Presentation:** 3
**Contribution:** 2
**Rating:** 4
**Confidence:** 3

**Summary:**

The authors identify residual connections as a powerful mechanism for provably avoiding negative transfer, and based on this insight, propose a simple yet effective method termed Residual Feature Integration (REFINE). Theoretically, the authors formally justify this simple yet remarkably effective approach through a rigorous analysis. Empirically, the authors conduct extensive experiments across image, text, and tabular benchmarks, and demonstrate that REFINE consistently mitigates negative transfer under diverse scenarios, including distribution shifts, label noise, semantic perturbations, class imbalance, and multi-source transfer settings.

**Strengths:**

- The manuscript is well written, logically structured, and easy to follow.
- The authors provide theoretical analysis.
- According to the empirical evaluations, the proposed simple approach achieves good performance across image, text, and tabular benchmarks, and under challenging conditions such as distribution shift, label noise, semantic perturbation, and class imbalance.

**Weaknesses:**

- The current setting is restricted to supervised fine-tuning. The proposed method requires access to labeled target-domain/task data, which is a relatively strong assumption even in conventional transfer learning [1]. Could the authors provide insights into whether the method would still work under unsupervised domain adaptation (UDA) [1] (does not require target-domain labels) or source-free domain adaptation (SFDA) settings [2-3] (does not require access to either the source data or target-domain labels)? Empirical validation or theoretical discussion in such scenarios would substantially enhance the generality of the method.
- The manuscript does not investigate how the performance varies with different amounts of target-domain data used for transferring.
- This manuscript lacks clarification of model architectures and size. The specific architectures of the CNN and Transformer models used in experiments are not clearly described. This makes it difficult to assess how architectural design and parameterization influence the effectiveness of the proposed method.
- For intentional and extreme negative transfer, such as non-transfer learning [4-6] (where the goal is to enforce strong negative transfer for model intellectual property protection), will the proposed method can still work? It would be insightful if the authors could discuss or empirically investigate the applicability or limitations of the method in such settings.

[1] Unsupervised domain adaptation by backpropagation. ICML 15\
[2] Do We Really Need to Access the Source Data? Source Hypothesis Transfer for Unsupervised Domain Adaptation. ICML 20\
[3] A Comprehensive Survey on Source-free Domain Adaptation. TPAMI 23\
[4] Non-Transferable Learning: A New Approach for Model Ownership Verification and Applicability Authorization. ICLR 22\
[5] Your Transferability Barrier is Fragile: Free-Lunch for Transferring the Non-Transferable Learning. CVPR 24\
[6] Toward Robust Non-Transferable Learning: A Survey and Benchmark. IJCAI 25

**Questions:**

Please see the weaknesses.

---

> ### Author Response · Authors · 2025-12-01
>
> We thank the reviewer for the helpful feedback and address the comments point by point below.
>
>
> - [Method is restricted to supervised fine-tuning, not Unsupervised Domain Adaptation (UDA) and Source-Free Domain Adaptation (SFDA).] Thank you for raising this important point. We agree that UDA and SFDA are valuable settings. Our work, however, intentionally focuses on supervised transfer learning because the core guarantee of REFINE relies on access to true target-domain labels. These labels are essential for controlling the residual term that drives our “no-worse-than-scratch’’ guarantee. In contrast, UDA and SFDA assume unlabeled target data, and theoretical analyses in those regimes typically require strong conditions such as  $P_s(Y|X) \approx P_t(Y|X)$ to avoid negative transfer. When this alignment fails, both UDA and SFDA methods can perform worse than scratch training, and no analogous safety guarantee is known. SFDA is even more fragile due to pseudo-label drift.
> Because of these fundamental differences, REFINE does not directly extend to UDA/SFDA: replacing true labels with pseudo-labels violates the conditions required for our safety analysis. Our contribution therefore addresses safe supervised transfer learning, whereas UDA and SFDA aim to leverage unlabeled data but cannot generally ensure non-degradation guarantees. Exploring whether residual-based correction can be adapted to unlabeled transfer regimes is an interesting direction for future work.
>
>
> - [Investigate how the performance varies with different amounts of target-domain data.] Thank you for the question. Our spatial-omics experiment has already examined how performance changes with the amount of target-domain supervision. As the number of labeled target cells increases from 1,000 to 3,000, accuracy improves from approximately 0.69 to 0.74 and macro-F1 from roughly 0.52 to 0.71. This trend is consistent with our design: additional labels allow the residual encoder to more accurately capture spatial structure omitted by the pretrained RNA-only model. Thus, this experiment provides empirical evidence of how performance scales with the amount of target-domain labeled data.
>
> - [The specific architectures of the CNN and Transformer models used in experiments.]
> Architectures are described in the submission and clarified in Sec. D. CNN finetuning uses a standard three-block convnet (32/64/64 channels; 3×3 conv + ReLU + 2×2 max-pool per block) followed by a 512-dim FC layer; the pretrained CNN backbone uses stages 80/160/320/640/640/768 with a 2560-dim projection head. Transformer finetuning uses patch size 4, embed dim 128, two encoder layers, and a 512-dim MLP head; the pretrained model uses patch size 2, embed dim 512, six encoder layers, and a 2560-dim projection head. For DomainNet, we use ResNet-10 for finetuning and ResNet-18 (torchvision) for pretraining. These are the exact architectures used in all experiments. Table S6 shows REFINE uses similar or fewer parameters than adapters (4.88% vs. 5.46% on CIFAR-CNN; 4.63% vs. 6.49% on CIFAR-Transformer). Fig. S2 further shows that increasing adapter capacity (1× to 500×) does not resolve negative transfer (accuracy stays around 65–66%), whereas REFINE reaches around 70.3% accuracy and 0.79 AUC, demonstrating that formulation—not capacity—eliminates negative transfer.
>
>
> - [Non-transfer learning uses a different setting.] Thank you for raising this point. Non-transfer learning (NTL) explicitly aims to induce strong negative transfer for model IP protection, whereas our setting is the opposite: we assume an honest user who seeks to recover good performance on the target domain. Under an adversarial or extremely misaligned target scenario (e.g., heavy label flips or semantic confusion), REFINE effectively acts as a “repair’’ mechanism. Because the residual $h(x)$ is trained using true target-domain labels, it learns whatever structure is missing from the corrupted source model, regardless of whether the corruption arises from accidental bias or deliberate NTL-style distortion. This behavior is reflected in Fig. 2 / Table 2: when the backbone is severely corrupted (e.g., 80% label flips or semantic confusion), most fine-tuning baselines collapse, whereas REFINE remains close to the clean NoTrans baseline and recovers target performance. In summary, REFINE is not intended to enforce NTL (it cannot create a protective barrier), but if a pretrained model has been intentionally “attacked,” an honest user with labeled target data can still use REFINE to undo the harmful drift.

---

### Author Response · Authors · 2025-12-02
**Summary of Major Reviewer Concerns and Our Responses**

Dear Area Chair and Reviewers,

For convenience, we summarize all major issues raised during review and how they are addressed in our responses and in the revised manuscript (changes highlighted in blue):

(1) Restriction to supervised fine-tuning (not UDA / SFDA).
We clarify that REFINE is intentionally designed for supervised transfer because its core safety guarantee relies on true target-domain labels; replacing these with pseudo-labels in UDA or SFDA violates our analytical assumptions, and extending to unlabeled regimes is left for future work.

(2) Performance versus target data size.
We note that our spatial-omics experiment already varies labeled target data from 1,000 to 3,000 samples and shows consistent improvements in accuracy and macro-F1 as supervision increases.

(3) Unclear model architectures.
We expand Section D to provide full specifications of all CNN, Transformer, and ResNet architectures used in experiments.

(4) Intentional negative transfer / NTL settings.
We clarify that REFINE does not enforce non-transfer learning barriers but acts as a repair mechanism that can correct severely corrupted pretrained representations when trained with true target labels.

(5) Justification of the no-negative-transfer theoretical guarantee.
We clarify the theoretical objective and add a new corollary directly establishing the guarantee relative to both baselines, along with an asymptotic result in the appendix.

(6) Tuning of hyperparameter $\rho$.
We clarify that $\rho$ is selected via standard validation or small grid search, using residual capacity comparable to a scratch model as a practical default.

(7) Bounded-norm assumption.
We remove this assumption, noting it was only introduced for notational simplicity.

(8) Validity for ERM solutions.
We explicitly prove in the revision that the ERM estimator satisfies the no-negative-transfer inequality under squared-loss assumptions.

(9) Relation to prior work (AMT and DTL).
We expand the related-work section to clarify REFINE’s representation-level adaptation, support for new modalities, and explicit safety guarantees, distinguishing it from prior output-level or retraining-based methods.

(10) Low CIFAR100 to CIFAR10 accuracy.
We clarify that the limited pretraining size is a deliberate choice for controlled stress testing, and we emphasize additional large-scale validation via spatial-omics experiments.

(11) Small- and medium-scale datasets.
We clarify that these datasets enable controlled analysis of negative transfer, while our guarantees are scale-agnostic and supported by foundation-model experiments.

(12) Choice of baselines.
We clarify that baselines were chosen to evaluate negative-transfer safety rather than peak performance, following standard NT practice.

(13) Parameter-efficiency comparisons.
We add detailed parameter analyses (Table S6 and Figs. S3-S4) showing that increasing adapter capacity alone does not mitigate negative transfer.

(14) Multi-source baselines.
We clarify that our multi-source experiments are conducted in a source-free adapt-time setting, making NoTrans and naive concatenation the only feasible baselines.

---

### Meta-Review · Area_Chair_GmPh · 2025-12-03

**Summary:**

The paper introduces ReFine, a method to mitigate negative transfer during fine-tuning. The approach is simple and easy to implement: it augments a frozen pre-trained model with a trainable residual encoder, allowing for correcting representations without destroying source knowledge. The authors supported this method with extensive validation across diverse settings. Furthermore, the submission offers theoretical guarantees in the form of generalisation bounds, showing that the method performs no worse than training from scratch. While the reviewers initially raised significant criticisms regarding both the theoretical proofs and empirical settings, the authors provided a comprehensive rebuttal. Unfortunately, the authors posted their response late (after the OpenReview incident), which prevented the reviewers from engaging. I am therefore forced to guess how the discussion would have evolved and I am inclined to believe that the rebuttal addresses the main critiques, and the work merits acceptance.

**Reviewer Concerns:**

The reviewers' concerns spanned theoretical novelty, scope, and empirical rigour.

Regarding theory, Reviewer PRLw argued that the main theorem did not support the claim of preventing negative transfer. The authors addressed this by adding Corollary 4.4 and Proposition A.1 to the revision.

Regarding the scope, Reviewer VFA3 criticised the restriction to supervised fine-tuning, suggesting the inclusion of UDA and SFDA. Unfortunately, this is a strict requirement in their safety guarantees that rely on true target labels to control the residual term.

On the empirical side, Reviewer fQBT raised concerns about sub-standard performance on CIFAR benchmarks. The authors clarified that this was a deliberate design choice involving a "lightweight" setup to perform controlled stress tests. Similarly, Reviewer cD6H's critique regarding "outdated" baselines was addressed by clarifying that methods like DANN and LoRA remain the standard for evaluating safety in negative transfer literature.

**Reviewer Scores:**

What the reviewers would have done if able to participate fully in the discussion following the rebuttal is very hard to tell, given the absence of interaction. Based on the rebuttal, I would have expected a positive shift in their assessment.

- Reviewer VFA3 (Score: 4) might have raised their score marginally because of the standing limitations with UDA and SFDA.
- Reviewer PRLw (Score: 2) would likely have changed their recommendation to marginal acceptance, as their primary reason for rejection (the missing proof for the negative transfer guarantee) was resolved by the new corollaries in the revision.
- Reviewer fQBT (Score: 4) probably would have increased their score as well, given that their hesitation regarding "low performance" came from a misunderstanding of the experimental design. However, I wouldn't be surprise if the reviewer was to push back on some of the points.
- Finally, Reviewer cD6H (Score: 2) might have improved their score but remained skeptical; while their concerns about parameter efficiency were addressed with new analyses, their fundamental view that the theoretical contribution resembles standard approximation theory might persist.

---

### Decision · Program_Chairs · 2026-01-26

Accept (Poster)